



# Atmospheric VOC measurements at a High Arctic site: characteristics and source apportionment

Jakob B. Pernov[1], Rossana Bossi[1], Thibaut Lebourgeois[2], Jacob K. Nøjgaard[1], Rupert Holzinger[3], Jens L. Hjorth[1], Henrik Skov[1]

[1]Department of Environmental Science, iCLIMATE, Aarhus University, Roskilde, Denmark
    [2] Ecole Normale Supérieure, Department of Geosciences, PSL Research University, Paris, France
    [3] Institute for Marine and Atmospheric Research, Utrecht University, The Netherlands

*Correspondence to*: Jakob Boyd Pernov (jbp@envs.au.dk) and Rossana Bossi (rbo@envs.au.dk)





**Abstract.** There are few long-term datasets of volatile organic compounds (VOCs) in the High Arctic. Furthermore, knowledge
about their source regions remains lacking. To address this matter, we report a long-term dataset of highly time-resolved VOC
measurements in the High Arctic from April to October 2018. We have utilized a combination of measurement and modeling
techniques to characterize the mixing ratios, temporal patterns, and sources of VOCs at Villum Research Station at Station
Nord, in Northeast Greenland. Atmospheric VOCs were measured using Proton Transfer-Time of Flight-Mass Spectrometry
(PTR-ToF-MS). Ten ions were selected for source apportionment with the receptor model, positive matrix factorization (PMF).
A four-factor solution to the PMF model was deemed optimal. The factors identified were Biomass Burning, Marine
Cryosphere, Background, and Arctic Haze. The Biomass Burning factor described the variation of acetonitrile and benzene.
Back trajectory analysis indicated the influence of active fires in North America and Eurasia. The Marine Cryosphere factor
was comprised of carboxylic acids (formic, acetic, and propionic acid) as well as dimethyl sulfide (DMS). This factor displayed
a clear diurnal profile during periods of snow and sea ice melt. Back trajectories showed that the source regions for this factor
were the coasts around North Greenland and the Arctic Ocean. The Background factor was temporally ubiquitous, with a slight
decrease in the summer. This factor was not driven by any individual chemical species. The Arctic Haze factor was dominated
by benzene with contributions from oxygenated VOCs. This factor exhibited a maximum in the spring and minima during the
summer and autumn. This temporal pattern and species profile are indicative of anthropogenic sources in the mid-latitudes.
This study provides seasonal characteristics and sources of VOCs and can help elucidate the processes affecting the
atmospheric chemistry and biogeochemical feedback mechanisms in the High Arctic.

# 1 Introduction

The temperature in the Arctic has increased at twice the speed of the global average (IPCC, 2019), a phenomenon known as
Arctic amplification. Increased $CO_2$ concentration and sea ice loss are responsible for the majority of this temperature increase
(Dai et al., 2019). However, short-lived climate forcers (SLCFs; methane, ozone, black carbon (BC) and aerosol particles) are
together responsible for half of the present temperature increase observed in the Arctic (Quinn et al., 2008). Atmospheric
aerosol particles are the most important SLCF (due to their scattering, absorbing, and cloud modification properties) but their
climate forcing is associated with the largest uncertainty, especially in the Arctic (Pörtner, 2019). Ozone is an important
photochemical oxidant in the Arctic troposphere. Ozone precursors, e.g., VOCs, $NO_x$, and peroxyacetyl nitrate (PAN),
remained poorly characterized in the High Arctic (AMAP, 2015). Photochemical reactions including ozone and VOCs have
important implications for the lifetime of methane, a major greenhouse gas. The identification and characterization of processes
leading to precursor emissions of aerosols and ozone are therefore needed to improve the assessments of biosphere-aerosol-
climate feedback mechanisms.

In the Arctic, there are strong seasonal variations in aerosol size and concentrations, with long-range transport of
accumulation mode particles in late winter and spring and local production of ultra-fine particles in the summer and autumn
season (Flyger et al., 1980; Barrie et al., 1981; Heidam et al., 2004; Nguyen et al., 2016). Expansion of the polar dome and





inefficient wet removal in the winter and spring allows for the transport of anthropogenic pollution (sulfate aerosols with acidic and toxic components, BC, and VOCs) to the Arctic (Klonecki et al., 2003; Heidam et al., 2004; Nielsen et al., 2019). Several studies have reported on new particle formation (NPF) events, involving naturally emitted biogenic VOCs during the summer. Dall'Osto et al. (2018b) recently demonstrated a negative correlation of NPF events at Villum Research Station, Station Nord,

in North Greenland with sea ice extent. The authors suggested that ultrafine aerosol formation is likely to increase in the future,– given the projected increased melting of sea ice (Boe et al., 2009; Bi et al., 2018). Dall Osto et al. (2017) hypothesized that NPF events during summer on Svalbard were linked to marine biological activities within the open leads and between the pack ice and/or along the marginal sea-ice zones. Further confirming the same processes are occurring for Northeast Greenland (Dall'Osto et al., 2018a; Nielsen et al., 2019). Open leads and open pack ice emit DMS that undergoes atmospheric oxidation

leading to methanesulfonic acid (MSA), sulfur dioxide, and ultimately sulfuric acid, which helps form and grow particles (Nielsen et al., 2019). After formation, aerosols grow to sizes where they can act as cloud condensation nuclei (CCN) (Ramanathan et al., 2001). VOCs of marine biogenic origin greatly contribute to CCN activity during summer (Lange et al., 2018; Lange et al., 2019). The sources of NPF in the Arctic and its corresponding precursors are a topic of intense research, as uncertainty remains regarding the mechanism of aerosol production. For example, Burkart et al. (2017) found that the

condensable vapors responsible for particle growth were more semi-volatile than previously observed in mid-latitudes, although they could not identify a source area for these vapors. Aerosol formation is one of the most important factors in determining the surface energy balance in the Arctic. Recently, it was estimated that NPF events could increase CCN concentrations by 2–5 fold over background concentrations (Kecorius et al., 2019). However, parametrization of the processes leading to aerosol formation is still a large source of uncertainty in global radiative forcing predictions (Haywood and Boucher,

2000). The characterization of these gas-phase precursors to particle formation is a key factor for understanding the dynamics of the Arctic troposphere and the corresponding effects on climate.

Ozone has a distinct seasonality in the Arctic, with maximum mixing ratios in the winter, depletion events in the spring, and a minimum in the summer (Bottenheim and Chan, 2006). Ozone is an important pollutant at the surface and greenhouse gas in the mid to upper troposphere. Ozone can perturb radiation fluxes and modify heat transport to the Arctic

(Shindell, 2007). In the Arctic, sources of ozone include long-range transport and photochemical production. Ozone and its precursors (VOCs, $NO_x$, CO, and PAN) can be transported from anthropogenic sources in the mid-latitudes (Hirdman et al., 2009) and natural boreal forest fire emissions (Arnold et al., 2015), which have been increasing in recent years (Parrish et al., 2012). The major sink for ozone in the Arctic is photochemical reactions, followed by minor contributions from dry deposition. Ozone largely controls the oxidative capacity of the atmosphere, as a chief precursor for OH, an oxidant for many compounds,

and a major prerequisite for halogen explosion event (Simpson et al., 2007). Halogen explosion events can affect the lifetime and reaction rates for organic gases and the deposition of mercury to the Arctic ecosystem. Formaldehyde, a product of photochemical degradation and ozonolysis of VOCs, is also an important photolytic source of OH radicals particularly at high solar zenith angles, i.e., Arctic summer (Cooke et al., 2010). Photochemical reactions involving VOCs can be a sink (by reactions with ozone) and a source (through reactions with $NO_x$) of ozone. Increased anthropogenic activity in the Arctic





(shipping and resource extraction) is expected to increase emissions of both $NO_x$ and VOCs (Law et al., 2017). Biomass burning emissions, which are expected to increase in the future, have been shown to increase ozone production by as high as 22 % in the Arctic (Arnold et al., 2015). Ozone levels have consequences for OH radical production, which is the main oxidant of methane, thus largely controlling its lifetime in the atmosphere. Therefore, the characterization of the interactions of ozone and VOCs have implications for climate effects and atmospheric chemistry.

Several factors, including chemical lifetime, local emissions, and long-range transport, govern mixing ratios of VOCs in the Arctic atmosphere. The chemical lifetime of most VOCs in the Arctic is dependent on the oxidative capacity of the atmosphere, thus there is a strong seasonality (Gautrois et al., 2003). However, due to the low humidity in the Arctic atmosphere, the concentration of OH is low (Spivakovsky et al., 2000). Therefore, halogen and ozone chemistry plays an active role during the spring in the atmospheric chemistry of VOCs in Arctic regions (Simpson et al., 2015). However, atmospheric

reactions alone seem unable to explain the VOCs mixing ratios and dynamics observed at Arctic sites (Grannas et al., 2002; Guimbaud et al., 2002; Sumner et al., 2002), indicating missing sources other than photochemical production. Two potential local sources are the snowpack and the sea surface microlayer. The snowpack also has a major impact on ambient VOCs by uptake/release mechanisms and acts as a matrix for many photochemical and biological processes (Guimbaud et al., 2002; Grannas et al., 2004; Kos et al., 2014). For example, Dibb and Arsenault (2002) demonstrated that the snowpack is a source

of formic and acetic acid through the oxidation of ubiquitous organic matter. Furthermore, Boudries et al. (2002) observed emission from the snowpack to the atmosphere of acetone, acetaldehyde, and formaldehyde, which were explained by photochemical production in the snowpack and depositional fluxes of methanol was also observed, which they postulated as a source of formaldehyde. These observed gas-phase fluxes had a diurnal cycle following polar sunrise that correlated with the solar zenith angle. Sea surface microlayer emissions are important local sources of atmospheric VOCs e.g. DMS, formic acid,

and acetic acid (Mungall et al., 2017). Sea emissions have a pronounced seasonality because of sea ice preventing air-sea exchange during most of the year in the Arctic. The sea surface microlayer could play a role in the emission of VOCs due to photochemical processes (Chiu et al., 2017; Bruggemann et al., 2018) or heterogenic oxidation (Zhou et al., 2014). For highly water-soluble compounds, the ocean could also be an important sink (Sjostedt et al., 2012). Finally, transport of VOCs, such as benzene, methane, ethane, propane, and chlorofluorocarbons, has been observed from the mid-latitudes to the High Arctic

(Stohl, 2006; Harrigan et al., 2011; Willis et al., 2018).

Few studies have reported VOCs in ambient air from Arctic sites with on-line techniques, usually during short-term campaigns. Hornbrook et al. (2016) utilized non-methane hydrocarbons measurements to derive time-integrated halogen mixing ratios during the OASIS-2009 campaign at Barrow, AK. Mungall et al. (2018) studied the sources of formic and acetic acid at Alert, CA during the summer of 2016. Sjostedt et al. (2012) and Mungall et al. (2017) performed VOC measurements

onboard the CCGS *Amundsen* in the Canadian Archipelago during the summer of 2008 and 2014, respectively. There have been several campaigns exploring snowpack emissions of VOCs (Boudries et al., 2002; Dibb and Arsenault, 2002; Guimbaud et al., 2002; Barret et al., 2011; Gao et al., 2012). Gautrois et al. (2003) reported long-term VOC concentrations for Alert, CA, where a seven-year time-series of VOCs mixing ratios has been generated although with low time resolution, off-line



techniques (GC-MS). These previous studies call for higher time resolved and longer measurement campaigns, thus
highlighting the importance of long-term high time-resolved measurements of VOCs in the Arctic.

In this study, we report several months of high time-resolved mixing ratios of selected VOCs measured at the high
Arctic site Villum Research Station (Villum) at Station Nord (North Greenland). This study aims to provide a better insight
into the dynamics, seasonal behavior, and potential sources of VOCs in the high Arctic. We accomplish this by combining
VOC mixing ratios with meteorological data, air mass back trajectories, and the receptor model, positive matrix factorization
(PMF). In Sect. 2, we describe our analytical instrumentation and models in detail. In Sect. 3, we cover the seasonal dynamics
of VOC as well as each factor from the PMF model.

## 2 Methods

### 2.1 Field site

The sampling campaign took place at Villum Research Station (Villum), which is situated on the Danish military base, Station
Nord, in Northeastern Greenland (81° 36' N, 16° 40' W, 24 m above mean sea level). Villum is situated in a region with a dry
and cold climate where the annual precipitation is 188 mm and the annual mean temperature is -16 °C. The dominating wind
direction is southwestern with an average wind speed of 4 m sec$^{-1}$. The sampling took place about 2.5 km south-west of the
main facilities of the Station Nord military camp. The sampling location is upwind from the Station most of the time (Fig. S1).
An overview of the meteorological data is presented in Fig. S2. Statistics for meteorological data over the sampling campaign
can be found in Table S1.

### 2.2 Gas-phase measurements and data processing

Gas-phase measurements of VOCs were obtained using a PTR-ToF-MS 1000 (Ionicon Analytik GmbH). The measurement
campaign commenced after polar sunrise on April 4 and concluded before polar sunset on October 28, 2018. The PTR-ToF-
MS was operated with hydronium ion ($H_3O^+$) as a reagent ion, a drift tube temperature of 70 °C, a drift pressure of 2.80 mbar,
a drift tube voltage of 650 V leading to an *E/N* (Electric field/density of the buffer gas in the drift tube) value of around 120
Townsend (Td). Mass spectra up to *m/z*=430 Da were collected at 5 seconds scan rate. The instrument inlet consisted of a
PEEK capillary tube heated at 70 °C and a built-in permeation unit (PerMasCal; Ionicon Analytik) which emitted 1,3-
diiodobenzene, used for mass scale calibration. The inlet of the sampling line consisted of ¼" Teflon tubing extending through
an insulated opening in the roof with a sampling cone at the tip to prevent water and debris from blocking the orifice. Ambient
outdoor air was aspirated into the instrument at a rate of 100 ml min$^{-1}$. Blank measurements were obtained every 4 hours for
15 minutes by automatic switching from the ambient outdoor air to indoor air pumped through a Zero Air Generator (Parker-
Balston, Part #75–83). Due to technical issues (mainly electrical power failure), measurements were interrupted for short
periods ranging from days to weeks in April, June, August, and September. Instrument parameters (E/N ratio, drift tube





temperature, pressure, and voltage) were inspected before and after power failures to ensure proper instrument functionality.
Periods with abnormal parameter values were removed.

Data generated by the PTR-ToF-MS instrument were processed with the PTR-MS Viewer software v. 3.2.12 (Ionicon Analytik). Mass calibrations and VOC mixing ratios were calculated by the PTR-MS Viewer. The instrument quantification was validated against an external gas-phase calibration standard (Apel-Riemer Environmental), a comparison between standard and instrument mixing ratios yielded percent errors that were within the analytical uncertainties, therefore we are confident in
the quantification method. Inspection of the mass spectrum yielded nine protonated masses from which an empirical formula was calculated, and a compound name assigned. Compound names were assigned based on comparison with the libraries from the PTR-MS Viewer and Pagonis et al. (2019), and references therein, as well as *a priori* knowledge. For one compound, an empirical formula was calculated ($C_5H_8OH^+$) but a compound name could not be assigned due to the inability of PTR-MS to distinguish isomers. Another compound ($C_4H_8OH^+$) was doubly assigned to propionic acid and methyl acetate. Methyl acetate
has the same protonated m/z ratio as propionic acid; thus, contributions of methyl acetate to the signal at 75.058 m/z cannot be ruled out. Due to a proton affinity below water, saturated hydrocarbons (alkanes) are unable to be detected via PTR-ToF-MS. See Table 1 for a detailed list of selected masses. Output files were further processed with MATLAB R2018B for time averaging and blank values subtraction. The limit of detection (LOD) for each identified species was calculated as three times the standard deviation (s.d.) of the blank values for each day. For calculation of statistics, mixing ratios below LOD was set to
½ the LOD. The data was time-averaged to a 30-minute mean. The data set has been rigorously quality controlled, through analysis of particle number size distributions (PNSD), meteorological data (wind direction and speed), and internal activity logs, to remove the influence of local pollution. Uncertainty in VOC measurements accounted for the reaction rate coefficient as well as primary ion counts and blank corrected ion counts, for a detailed description see the Supplement. Ozone ($O_3$) was measured using an API photometric $O_3$ analyzer M400, data is quality assured and controlled via standard EN14625:2012,
with calibrations every six months (Skov et al., 2004; Skov et al., 2020).

**2.3 Positive Matrix Factorization (PMF) analysis**

The PMF model was operated using the US EPA PMF version 5.0 software, which uses the second version of the multilinear-engine 2 (ME-2) platform (Paatero and Tapper, 1994). The goal of PMF is to identify the number of factors or sources *p*, the species profile *f*, and the mass contributed by each factor to each sample. PMF accomplishes this by decomposing a data matrix
*X* into two matrices *G* and *F*. The input data matrix *X* consists of dimensions *i* and *j*, where *i* is the number of samples and *j* is the measured chemical species. The source profile matrix *f* is of dimensions *p* and *j*. The source contribution matrix *g* is composed of *p* and *i* dimensions. This is expressed in Eq. (1), below.

$$X_{ij} = \sum_{k=1}^{p} g_{ik} \times f_{kj} + e_{ij} \tag{1}$$

Where $e_{ij}$ is the residual matrix and *k* are the individual sources. PMF uses measurement uncertainties $u_{ij}$ and the residual
matrix to minimize the Objective Function *Q*, Eq. (2) below:





$$Q = \sum_{i=1}^{n} \sum_{j=1}^{m} \left[\frac{e_{ij}}{u_{ij}}\right]^2 \qquad (2)$$

Where $n$ is the total number of samples and m is the total number of species. There are three versions of the objective function: $Q_{true}$ that includes all data points, $Q_{robust}$ that excludes outliers, and $Q_{theo}$ that is approximately equal to the number of degrees of freedom. The ME-2 platform performs iterations via the conjugate gradient method until convergence to minimize $Q$. Each

good data point contributes a value of approximately one to the value of $Q$; therefore, $Q$ and the ratio of $Q_{true}$ to $Q_{theo}$ are the goodness of fit parameters for the appropriate number of factors (Paatero et al., 2014).

     The following data preparation protocol was developed according to standard practice in the field (Polissar et al., 1998; Reff et al., 2007; Hopke, 2016) which allows PMF analysis to be performed effectively. In certain cases, discussed here, the data set was modified before modeling via PMF. Data with concentrations below the LOD were replaced with a value

equal to half of the LOD. The associated uncertainty was set to 5/6 of the LOD. Missing concentrations from a sample were replaced with the median concentration of the data set and the uncertainty was set as a multiple (3) of median concentration (Polissar et al., 1998; Reff et al., 2007). It is worth noting that the operational protocols used to estimate the uncertainties and treatment of data are based on extensive testing to find an approach that provided useful results (Hopke, 2016).

     Two methods for evaluating modeling uncertainty in PMF were performed: bootstrapping (BS) and displacement of

factor elements (DISP) (for a description see Paatero et al. (2014)). BS uncertainty includes effects from random errors and partially includes the effects of rotational ambiguity. DISP explicitly captures uncertainty from rotational ambiguity (Brown et al., 2015). Another method of estimating rotational ambiguity is the Fpeak function. Fpeak evaluates $Q$ under different rotational strengths, in this study Fpeak strengths from -5 to 5 in intervals of 1 and from -1 to 1 in intervals of 0.1.

### 2.4 Ancillary data

Meteorological data including temperature, relative humidity, wind speed, wind direction, pressure, radiation, and snow depth were generated by an automatic weather station placed close to the measurement site. Using the local wind direction and wind speed, a conditional probability function (CPF) was calculated using the source contributions for each factor. CPF is defined as CPF $= m_\theta/n_\theta$, where $m_\theta$ is the number of occurrences that a source contribution exceeds a predetermined threshold criterion (75[th] percentile) while arriving form a wind sector and $n_\theta$ is the total number of occurrences wind arrived from the same wind

sector. A wind sector was defined as 30° and wind speeds below 0.5 m s[-1] were excluded to account for uncertainty in wind direction at low wind speeds. Daily polar gridded sea ice concentrations for the measurement period were obtained through the Nimbus–7 SMMR and DMSP SSM/I-SSMIS Passive Microwave Data (Cavalieri et al., 1996). Time series of local sea ice concentrations were calculated from the gridded daily average sea ice concentrations (%) by masking an area of ± 2° longitude and +8°/-4° latitude around Villum (Greene et al., 2017; Greene, 2020). Active fires during the period, August 15–September

15, 2018, was provided by NASA's Fire Information for Resource Management System (FIRMS) which distributes Near Real-Time (NRT) active fire data from NASA's Moderate Resolution Imaging Spectroradiometer (MODIS) and NASA's Visible Infrared Imaging Radiometer Suite (VIIRS) (Schroeder et al., 2014).





### 2.5 Back Trajectory Analysis

The back trajectory model HYSPLIT (Draxler and Hess, 1998; Rolph et al., 2017) calculated air mass back trajectories arriving
at Villum, incorporating Global Data Assimilation System (GDAS) meteorological data with 1° spatial resolution, and
employing modeled vertical velocity. Air mass back trajectories were calculated at 100 m arrival altitude. The trajectory length
was varied between 240 and 336 hours. For a synoptic view of air mass history, trajectory frequency maps were calculated
following a similar methodology utilized by Tunved et al. (2013) and Freud et al. (2017). Grids of $1° \times 1°$ cells were centered
concentrically on Villum; the number of individual trajectories passing over each grid cell was summed, normalized by the
total number of trajectories, and multiplied by 100 % to give a trajectory frequency map. The large number of trajectories
included in the frequency maps provide statistical robustness to their interpretation and reduces their associated uncertainty.

### 3 Results and discussion

### 3.1 VOC temporal patterns and mixing ratios

For the ten selected VOCs, time series of mixing ratios during the entire measuring period are displayed in Fig. 1 (a-f). Details
for each compound are presented in Table 1. During the spring (April–May), certain compounds (benzene and $C_5H_8O^+$)
exhibited a maximum and thereafter a decreasing pattern, similar to the timing and profile of the Arctic Haze phenomena;
whilst in summer (June–August) certain compounds (DMS and OVOCs) revealed a diurnal cycle that closely follows radiation.
Interestingly, several compounds (formaldehyde, formic acid, and acetone) peaked in the spring with decreasing levels until
the summer when a diurnal pattern following sunlight was observed. The levels, seasonal patterns, and comparison with other
studies of these compounds will be discussed below.

Oxygenated volatile organic compounds (OVOCs) selected for this study included formaldehyde, formic acid, acetic
acid, propionic acid, MEK (methyl ethyl ketone) and an ion with empirical formula $C_5H_8OH^+$ (possible isomers for this
compound include, among others, methyl butenal, pentenal, methyl butanone). Formaldehyde, formic acid, MEK, and acetone,
and to a lesser extent acetic acid and $C_5H_8OH^+$, displayed a decreasing pattern in the spring. For formaldehyde, formic acid,
acetic acid, acetone, MEK, and propionic acid a clear diurnal variation was observed in the period July–August, with peak
mixing ratios occurring around midday (Fig. 1 a, c, d, e). The diurnal variation was less pronounced in April–May and
September–October, highlighting the dependence on sunlight. Acetone showed the highest mean mixing ratio ± s.d. (0.608 ±
0.196 ppbv). Mean mixing ratios of acetone measured at Barrow, AK during the OASIS-2009 field campaign (March–April
2009) were 0.900 ± 0.300 ppbv (range of 0.364–2.21 ppbv) (Hornbrook et al., 2016), and in the Canadian Archipelago in
August–October was 0.424 ppbv (Sjostedt et al., 2012), which is within the same range observed at Villum (0.608 ± 0.196
ppbv, Table 1). The average mixing ratio of formaldehyde in the present study (0.220 ± 0.128 ppbv) is similar to those
measured at Barrow, AK (0.204 ppbv) and Alert, CA (0.166 ppbv) in March–April (Grannas et al., 2002; Barret et al., 2011).
Formic acid (0.454 ± 0.371 ppbv) and acetic acid (0.201 ± 0.149 ppbv) mean mixing ratios were within the range of those



measured at Summit, Greenland (0.4 ppbv) by Dibb and Arsenault (2002), although considerably lower than those measured by Mungall et al. (2018) during the early summer at Alert, CA (formic acid $1.23 \pm 0.63$ ppbv, acetic acid $1.13 \pm 1.54$ ppbv). MEK (an oxidation product of *n*-butane) displayed a mean mixing ratio of $0.031 \pm 0.021$ ppbv, which is slightly lower than the median concentrations of 0.190 ppbv measured in March–April 2009 at Barrow, AK (Hornbrook et al., 2016) and 0.054 ppbv measured at Alert, CA in April–May 2000 (Boudries et al., 2002).

The two main non-oxygenated compounds measured were acetonitrile and benzene. Benzene mixing ratios followed

the expansion of the polar dome with high mixing ratios in the spring period and lowest in the summer period (Fig. 1 f), similar to sulfate and BC measured (Massling et al., 2015; Skov et al., 2016) and accumulation mode aerosols (Lange et al., 2018). The mean mixing ratio of benzene measured at Villum was $0.027 \pm 0.016$ ppbv, which is a factor of two higher than those measured in the Canadian Archipelago (0.013 ppbv) by Sjostedt et al. (2012). Benzene has shown a seasonal pattern at Alert, CA with a higher mixing ratio in winter due to no or limited photochemistry and long-range transport from lower latitudes

(Gautrois et al., 2003). They reported mean winter and summer mixing ratios of 0.200 and 0.034 ppbv, respectively; when compared to the present study is a factor of two higher in the winter but in good agreement during the summer. Acetonitrile followed a similar pattern to benzene during the spring indicating a slight influence from anthropogenic emissions, minima in the summer, and maxima during the autumn (Fig. 1 b). The mean mixing ratio of acetonitrile observed at Villum is $0.067 \pm 0.025$ ppbv, which is a factor of two higher than reported by Sjostedt et al. (2012) (0.030 ppb). The range of acetonitrile mixing

ratios (0.023–0.156 ppbv) corresponds to the upper and lower limits of background levels over the Atlantic Ocean (0.10–0.15 ppbv) reported by Hamm et al. (1984) and Hamm and Warneck (1990).

DMS was the only sulfur-containing compound detected, with mean $\pm$ s.d. of $0.046 \pm 0.043$ ppbv. The mixing ratios of DMS observed in this study are a factor of two lower than those reported by Sjostedt et al. (2012) (0.093 ppbv). DMS mixing ratios were near LOD during the spring and autumn, however, were significantly elevated levels during the summer

(Fig. 1 e). DMS showed a clear diurnal cycle during sea ice melt in the summer months correlating with sunlight intensity.

**3.2 Springtime VOC correlations**

Elevated DMS mixing ratios were observed for two short periods of a few days' duration in May (May 1–5 and May 16–19). (See Fig. 2 Left and Right below). In May, most of the ocean surrounding Villum is still frozen. However, satellite images from the area (Fig. S4 a-f and S5 a-e) showed that there were open leads in the frozen sea surface and back trajectory

calculations (Fig. S6 a and b) confirmed that, during the DMS emission episodes, the air masses experienced extensive surface contact, traversed over the open leads before reaching the station. During DMS emission episodes, the acetone mixing ratios decreased correspondingly. Sjostedt et al. (2012) found moderate anti-correlation (R=0.37, p<10⁻⁴) for DMS with acetone. Minimum values of acetone were observed when DMS reached its maximum values, and the short photochemical lifetime of DMS suggests a localized biological sink for acetone associated with the production of DMS. Certain microorganisms can

consume acetone as well as produce DMS from DMSP (Taylor et al., 1980; Kiene et al., 2000). At Villum, the relationship between acetone and DMS showed seasonal changes with a moderate negative correlation in April (R=-0.55), a weak positive





correlation in July (R=0.23), and a strong negative correlation in September (R=-0.68). Possible reasons for these variations may be changes in the biological conditions of the seawater, photochemical activity, and sources regions. Pearson correlation coefficients for chemical species, radiation, and temperature for April, July, and September are tabulated in Table S2, S3, and
S4, respectively.

As illustrated in Fig. 2 a and b, acetone is anti-correlated with ozone during periods of elevated DMS. This relationship is particularly clear during situations with abrupt changes in the mixing ratios of the species as on May 1, 2, and 5. These changes in mixing ratios are accompanied by a change in meteorological conditions, illustrated here by changes in wind speed. Guimbaud et al. (2002) found a similar relationship between acetone and ozone during different field campaigns at Alert,
Canada with acetone increased during ozone depletion episodes accompanied by a concomitant decrease in the propane mixing ratios. However, it was found that the increase in acetone could not be explained by gas-phase chemistry but possibly by photochemically induced emissions from the snowpack. This phenomenon was also observed by Boudries et al, (2002). The anti-correlation between ozone and acetone observed at Villum may also be explained by a similar influence of photochemistry that causes destruction of ozone as well as the formation of acetone by gas phase and surface reactions. Also, the possible
influence of vertical air exchange must be considered as well. During pristine atmospheric conditions at Villum, ozone is destroyed but not produced within the boundary layer, due to low $NO_x$ concentrations (Nguyen et al., 2016). Exchange with the free troposphere will lead to increases in the ozone concentrations and possibly a reduction of acetone concentrations at ground level due to dilution by air from aloft with a lower acetone concentration. The anti-correlation between ozone and acetone supports the hypothesis that acetone is not brought down from aloft to a significant extent but has surface or boundary
layer chemistry as its main source.

During the summer, the behavior of acetone is different. In addition to the previously mentioned dependence on the diurnal variations of sunlight, providing strong evidence of a local photochemical source, a positive correlation with ozone was observed. In June, an anti-correlation is still seen, but in July and August, the two species are correlated (R=0.69 for July and R=0.46 for August). The fact that ozone is also positively correlated to other OVOCs (particularly formaldehyde, R=0.86
for July) suggests that the correlation is due to the influence of transport of air containing ozone and acetone formed by the photochemical degradation of air pollutants. During the summer period, acetone is correlated with acetonitrile (R=0.73 for June–August), in September and October this correlation becomes very strong (R=0.96). Acetonitrile is considered an atmospheric tracer of biomass burning as the global budget of this compound is dominated by emissions from biomass burning (Holzinger et al., 2001). Thus, biomass burning and atmospheric degradation of biomass burning products seem to be an
important source of acetonitrile and acetone during this period. The correlation with ozone is also positive during these months, most likely because the photochemistry of biomass burning emissions is also a source of ozone brought to Villum. The different temporal patterns and correlations suggest the behavior and sources of VOCs in the Arctic are seasonally dependent. Therefore, a detailed, statistical investigation of the sources affecting VOC levels is warranted.




### 3.3 Source Apportionment via PMF

VOCs exhibited distinct temporal patterns that are seasonally dependent and suggest different processes contributing to ambient mixing ratios. Therefore, the source apportionment model, PMF, was employed to provide an in-depth examination of these VOC sources. The base model was executed 100 times with a random start seed. Species were categorized based on their signal-to-noise ratio (S/N), species with an S/N ≥ 1, 0.2 < S/N < 1, S/N < 0.2 were categorized a 'Strong', 'Weak', and 'Bad', respectively. The uncertainty of 'Weak' species was tripled, and 'Bad' species were excluded from the analysis. Two

species deviated from this categorization; benzene (S/N = 0.3) since it serves as a tracer for anthropogenic emissions from fossil fuel combustion and formic acid (S/N =1.0) since there was substantial variability of blank measurements in the spring. Rather than down weighting spring samples, the entire dataset for formic acid was down-weighted to minimize bias for the spring period. The species included in the analysis were those shown in Table 2. Expanded uncertainties for model input were estimated as described in the Sect. 1 of the Supplemental Information. The two periods of elevated DMS mixing ratios were

removed from the model input matrix since these periods were considered an anomaly compared to the rest of the measurement period (appearance of open leads, wind direction directly from these leads, and air masses with extensive surface contact). Therefore, these periods violated one of the assumptions of PMF; that sources do not change significantly over time or do so in a reproducible manner. The inclusion of these two periods did not improve model performance. Instead, we argue that their exclusion allows us to model the ambient behavior of VOCs void of episodic influence due to certain meteorological conditions

315         A four-factor solution was deemed optimal based on $Q_{true}/Q_{theo}$ ratios, $R^2$ values between modeled and measured mixing ratios, and physical interpretation of the factor time series and profiles. Figure S7 displays the $Q_{true}/Q_{theo}$ ratios against the factor number. Increasing the factor number from two to three produces the largest decrease in the $Q_{true}/Q_{theo}$ ratio, which is often taken as the optimal solution for the number of factors. However, the mean $R^2$ values for the 3-factor solution (0.8) were lower than for the four-factor solution (0.85) and the physical interpretation of the four-factor solution yielded more

robust analysis. Therefore, a four-factor solution was deemed optimal. The large discrepancy between $Q_{true}$ and $Q_{theo}$ can be explained by the large analytical uncertainties (32–64 %, Table 1), which is due to the extremely low mixing ratios observed, causing $Q_{true}$ to be small, the large number of samples which produces a large $Q_{theo}$, as well as co-variation in the species (see Sect. 3.1). While these uncertainties are high, they are reasonable for a kinetic quantification of organics at these instrument parameters and extremely low mixing ratios based on Holzinger et al. (2019).

325         Displacement on the four-factor solution yielded no errors in the model and zero factor swaps, illustrating the solution is valid and free of rotational ambiguity. Bootstrapping was performed for 100 runs and mapped >85 % of the boot factors to the base factor. This high percentage indicates the model solution is free of random error. Variations of the Fpeak strength consistently returned the lowest change in $Q$ at Fpeak = 0, indicating the model is free of rotational ambiguity. The inspection of G-space plots produced no visible correlations between factors. Together these error estimation methods show the model

solution is robust, valid, and free of random errors and rotational ambiguity.



Based on their chemical composition and their temporal variation the four factors were assigned to likely sources including Biomass Burning, Marine Cryosphere, Background, and Arctic Haze, which will be explained in detail below.

### 3.3.1 Biomass Burning Factor

The most prominent species in the profile of Biomass Burning factor is acetonitrile, explaining 63 % of the variation, and
benzene, explaining 33 % of the variation (Fig. 3). As mentioned above, acetonitrile is a characteristic tracer for biomass burning emissions. Biomass burning is also an important source of benzene, with an estimated global strength of about half of the anthropogenic sources (Lewis et al., 2013) and it is a source of methyl acetate as well (Andreae, 2019). The chemical species profile (Fig. 3, bottom) of this factor, therefore, points to a biomass-burning source. The time series (Fig. 3, top), shows this factor to decrease in the spring to a minimum in the summer, and slowly increase to a maximum at the beginning of
September. The decrease in the spring is reflective of anthropogenic emissions of acetonitrile and benzene during this period as the polar dome is expanded allowing for emissions to be entrained from the mid-latitudes. The height of the biomass burning season in North America and Northern Eurasia is July (Lavoue et al., 2000), although due to the contraction of the polar dome during summer, minimum contributions from this factor are observed. Increased areas of open water in the Arctic also act as a sink during the summer (de Gouw et al., 2003). The Biomass Burning factor peaks in August/September when the polar
dome starts to expand thus allowing biomass burning emissions to reach the High Arctic.

While the species profile and temporal nature indicate biomass burning emissions as the source of this factor, where do these emissions originate from? Stohl (2006) revealed three major pathways for transport of air masses into the Arctic: low-level transport followed by ascent, low-level transport, and ascent outside the Arctic followed by descent into the Arctic. Emissions from North America and Asia only enter the Arctic through the last pathway. To examine the geographical origin
of this factor, air mass back trajectories from the HYSPLIT model were calculated every hour during the peak of the Biomass Burning factor (August 15–September 15, 2018) for 100–meter arrival altitude and extending 336 hours backward in time. This analysis combined with active fire data from the FIRMS database indicates the influence of active fires in North America and Eurasia (Fig. 4).

While most air masses stayed within the Arctic, there is evidence of overlap between air mass history and active fires
during this period in North America and Eurasia (Fig. 4). The influence of biomass burning was observed at other High Arctic sites during this period. Lutsch et al. (2019) used FTIR measurements of CO, HCN, and ethane at several High Arctic sites coupled with aerosol optical depth data and the GEOS-Chem model to detect the influence of wildfires and attribute their sources. They observed fire-affected enhancements in the tropospheric CO column at Eureka, CA from September 9 to the 25, 2018, and at Thule, GL from August 24 through September 26, 2018. The GEOS-Chem simulated the source regions for the
fire affected enhancements in the tropospheric CO column measurements to be boreal forests in North America and Asia at both sites (Lutsch et al., 2019). These observations of biomass burning at other High Arctic sites are in good agreement with the Biomass Burning factor presented here, adding robustness to this factor assignment.



Biomass burning is known to be an important source of BC, and it has been estimated to account for about 35 % of the BC emissions in the Northern Hemisphere (Qi and Wang, 2019). Despite this, the observed time profile of BC (not shown) at Villum did not show an increase during the autumn of 2018. This is likely to be explained by the fact that the emissions from biomass burning sources have been transported over long distances with corresponding long transport time, as BC is removed much faster from the atmosphere than acetonitrile due to wet deposition. The atmospheric residence time of BC is below 5.5 days, according to a recent estimate (Lund et al., 2018), while that of acetonitrile is several months (de Gouw et al., 2003). Using meteorological parameters calculated along the trajectory path, for air masses arriving at 100 m altitude, the mean accumulated precipitation for the peak of the Biomass Burning factor was 14 millimeters (mm). Raut et al. (2017) used a combination of in situ observations from aircraft, satellite remote sensing, and modeling simulations to calculate the transport efficiency of BC during 2012. They concluded that the transport efficiency of BC was low (<30 %) when accumulated precipitation was large (5–10 mm). These previous observations combined with the accumulated precipitation data along each trajectory during the peak of the Biomass Burning factor support the lack of BC loading during this time. While biomass burning is a source of BC globally, which is expected to increase in the future (Westerling et al., 2006), the results presented here indicate meteorological parameters encountered during transport can play a role in the levels observed in the High Arctic atmosphere. While biomass-burning emissions may increase in the future, increased precipitation patterns might counterbalance this increase although more research is needed to elucidate the relationship between these feedback mechanisms.

### 3.3.2 Marine Cryosphere Factor

The Marine Cryosphere factor was characterized by formic acid, acetic acid, propionic acid, and DMS, explaining over 50 % of the variability of each of these compounds (Fig. 5, Bottom). The contribution of this factor is near zero in the spring and autumn and maxima during the summer months. This factor shows an enhanced diurnal variation with a clear correlation to sunlight during the summer months (Fig. 5, Top). The high content of DMS points to a marine origin of this factor, while carboxylic acids have been demonstrated to be emitted from the snowpack (Dibb and Arsenault, 2002). Analysis of snow depth and sea ice concentrations (± 2° longitude and +8°/-4° latitude area around Villum) illustrate the onset of this factor coincides with the snowmelt and sea ice decline. Therefore, a combination of marine and cryosphere sources appears to contribute to the species observed in this factor.

The sources of the organic acids are much less well characterized than those of DMS; in fact, model simulations have not been able to reproduce the mixing ratios of formic and acetic acid, particularly in the Arctic and northern mid-latitudes (Paulot et al., 2011; Mungall et al., 2018). As the lifetimes of formic acid and acetic acid against photochemical oxidation by reaction with the OH radical are relatively long (about 25 and 10 days, respectively, for [OH]=$10^6$ molecules/cm$^3$), dry and wet deposition are thought to be the main removal pathways (Seinfeld and Pandis, 2016). Estimated globally averaged atmospheric lifetimes against wet deposition for formic and acetic acid in the boundary layer is between 1 and 2 days, respectively (Paulot et al., 2011). Thus, it is unlikely that direct long-range transport plays a relevant role in determining the





mixing ratios of these species at Villum. Analysis of $C^{14}$ isotopes in formic and acetic acid in air and rainwater have shown that outside of urban and semi-urban areas the dominating (>80 %) source is modern carbon (Glasius et al., 2001). This analysis is consistent with model simulations showing that atmospheric oxidation of biogenic hydrocarbons is the largest source (Paulot et al., 2011; Millet et al., 2015). Even though vegetation in the High Arctic is sparse, contributions from precursor emissions

or direct emissions of formic acid and acetic acid from vegetation cannot be excluded, as discussed by Mungall et al. (2018). Emissions from the soil is also a possible but highly uncertain source of these species (Mungall et al., 2018). However, the Marine Cryosphere factor is largely absent when snow is completely melted, exposing the bare ground and vegetation to the atmosphere, thus soil emissions and vegetation are improbable sources of these compounds. Instead, enhancements in these species and this factor is observed during periods of snowmelt and sea ice melt.

405         A comparison of the contribution of the Marine Cryosphere Factor to sea ice concentration, calculated as described in Sect. 2.3, and snow depth can further shed light on the origin of this factor (Fig. 5, top). Periods of high contributions and clear diurnal pattern by the Marine Cryosphere factor starts on June 23, where the local sea ice concentration and snow depth are starting to decline. Diurnal patterns were observed during this period of melting. This continues until August 7, when the measurements were interrupted due to technical issues. When measurements resumed on August 16, the contribution from the

Marine Cryosphere factor had returned to the low levels found during springtime. Note that instrument parameters were monitored before and after interruptions to ensure proper functionality of the instrument, and periods that deviated from nominal values were removed. The Marine Cryosphere Factor appears not to be strongly dependent on the extension of the open sea, as sea ice concentrations/extensions reach a minimum, and consequently, the open sea area reaches a maximum by the beginning of September, but rather depends on active melting of snow and sea ice. Thus, it seems that emissions of VOCs

from melting snowpack and newly exposed sea ice areas could offer a viable explanation for the observed dependence of this source.

        Previous work has shown that emissions from the sea in the Arctic area can be caused by a surface microlayer enriched in organic substances that acts as a source of formic acid and other oxidized VOCs (Mungall et al., 2017). This occurs either via heterogeneous chemistry or by photochemically driven reactions within the surface layer (Vlasenko et al., 2010; Chiu et

al., 2017). Mungall et al. (2017) performed factor analysis of VOCs in the Canadian Archipelago finding four factors. One factor (Ocean factor; containing formic acid, isocyanic acid, and oxo-acids) was highly correlated with dissolved organic carbon (DOC), fluorescent chromophoric dissolved organic matter (fCDOM), and radiation. However, DMS was poorly correlated with this factor. They concluded the source to be photochemical or heterogeneous oxidation from sources on the sea surface microlayer. While formic and acetic acid, as well as the carbonyl compounds, show clear daily variations

correlating with radiation, as mentioned above, DMS shows a less clear correlation. The emission of DMS from the open ocean has been demonstrated to be dependent on horizontal wind speed (Bell et al., 2013). Although, the variation of the Marine Cryosphere Factors seems not to be driven mainly by the dependence on horizontal wind speed (Fig. S2.). Marine microorganisms produce DMS (Stefels et al., 2007; Levasseur, 2013), and given the distance of the measuring site from open water, it is proposed that the majority of DMS produced is already oxidized to MSA and other products when reaching the





station. MSA has been previously measured in the particle phase at Villum in February–May 2015 (Dall'Osto et al., 2018b; Nielsen et al., 2019).

Several studies have demonstrated the emission of VOCs from the snowpack; Gao et al. (2012) observed photo enhanced release of VOCs from both Arctic and mid-latitude snow; Grannas et al. (2002) obtained similar results by applying a box model to simulate observed emissions of carbonyl compounds from an Arctic surface layer at Alert. They found that

diel cycles of carbonyl compounds are impacted by snowpack exchange characterized by nighttime adsorptive uptake from the snowpack and the largest release around noon, similar to the observations in this study. Anderson et al. (2008) found a high concentration of water-soluble organic compounds (presumably mainly formic and acetic acid) in the surface layer of polar snow, and Dibb and Arsenault (2002) had measured levels well above 1 ppbv of formic and acetic acid in firn air. Gao et al. (2012) also observed enhanced release of acetone, formic acid and acetic acid from snow coinciding with radiation, which

they explained by oxidation of organic matter e.g., humic substances present within the snowpack, perhaps by photochemically produced OH radicals (Nguyen et al., 2014). This experimental evidence that Arctic snow and areas of open sea are a relevant source of VOC emissions adds credence to this factor assignment.

The spatial origin of the Marine Cryosphere factor was investigated using 240-hour HYSPLIT air mass back trajectories arriving at 100 m altitude. These trajectories and trajectory frequency maps were calculated as described in Sect.

2.4. Figure 6 displays the trajectory frequency map for air masses arriving every hour from June to August, corresponding to the period of maximum contribution from the Marine Cryosphere factor and diurnal variation. From Fig. 6, three areas of air-mass origin can be discerned. Air masses arrived from regions along the eastern and northern coast of Greenland as well as from the Arctic Ocean. Marginal ice zones (MIZ), defined as areas where fragmented sea ice encounters land, have been identified as a source region of biogenic activity leading to new particle production (Dall Osto et al., 2017). Kecorius et al.

(2019) recently identified two types of NPF events on a cruise in the waters surrounding Svalbard; one being more hygroscopic, which they hypothesized to arise from sulfuric acid, while another was less hygroscopic, likely with higher contributions from organic species. Both types of NPF events were observed to have originated from both MIZs around the northern coast of Greenland and the Arctic Ocean north of Svalbard. MIZs are therefore proposed as a major source area for the Marine Cryosphere factor.

The properties of the Marine Cryosphere factor (composition, temporal variation, and spatial origins) helps confirm the work of previous studies in the High Arctic. We propose this factor (although not necessarily these exact species) as responsible for the biogenic precursor emissions of particles observed in other studies (Nguyen et al., 2016; Burkart et al., 2017; Dall Osto et al., 2017; Freud et al., 2017; Dall'Osto et al., 2018a; Dall'Osto et al., 2018b; Dall'Osto et al., 2019; Nielsen et al., 2019). For example, Nguyen et al. (2016) identified the area southeast of Villum as having a high probability of observing

an NPF event when air masses originating from this sector. One of the source areas identified in Fig. 6 is southeast of Villum, and a CPF analysis, high contributions were observed when the wind direction was south of Villum (Fig. S8 a). While the species identified using this analytical technique might not be responsible for particle formation and growth, other high molecular weight compounds originating from the same sources could well be. Therefore, this factor has important climatic



implications, as sea ice and snowmelt are expected to start earlier due to warming temperatures. Increased contributions from
this factor can be expected, which will alter the CCN budget and occurrence in the summer and thus alter the radiative balance.

### 3.3.3 Background Factor

The Background factor explains the majority (>50 %) of the variation of acetone and the ion $C_5H_8O^+$ as well as 37 % of
formaldehyde. It explains approximately 30 % of the variation of acetonitrile and MEK, followed by minor (< 20 %) variations
of acetic acid, benzene, and propionic acid. Most of its components, particularly acetone and formaldehyde, are known to have
photochemical oxidation of precursor compounds in the atmosphere as an important source. The chemical profile of this factor
does not point to a specific, known source (Fig. 7, bottom). Its contributions start increasing in the middle of April and reach
a maximum by the end of the month (Fig. 7, top). The contributions decrease until the summer period, where a slight diurnal
profile, albeit weaker in magnitude when compared to the Marine Cryosphere factor, can be recognized. During the autumn,
contributions levels are similar to the summer period, however, the temporal pattern is quite similar to the one observed for
the Biomass Burning factor. The temporal correlations of the Background factor to the Marine Cryosphere and Biomass
Burning factor during their respective periods of peak contributions indicate this factor does not arise from one identifiable
source but rather from a myriad of sources, hence the assignment as a background factor. The species profile for the
Background Factor corresponds to mixing ratios of 0.355 ppbv for acetone, 0.090 ppbv for formaldehyde, and less than 0.050
ppbv for all other compounds. These mixing ratios can be interpreted as the background mixing ratios for these compounds in
the High Arctic.

The Background factor has its highest period of mean contributions during the spring when solar intensity increases
but before the emissions related to open sea or melting snow become relevant. This factor likely represents a source of VOCs
caused by the increasing rate of photochemical oxidation of liable organic carbon naturally present in the air and on surfaces.
Photo-oxidation of alkanes present in the air and deposited during the winter is a possible source of liable carbon (Boudries et
al., 2002; Guimbaud et al., 2002; Gao et al., 2012). For example, acetone (a major component of the Background Factor) is
primarily formed from reactions of OH and Cl with propane, isobutane, and pentane (Hornbrook et al., 2016). This slow
decrease during the spring could be due to the decreasing supply of liable carbon in the snowpack. The weak diurnal pattern
in the summer could be due to increased available organic matter for oxidation from the open ocean and melting snowpack.
Further measurements, especially during the polar night to day transition, are required to test this hypothesis.
Given the lack of a peak period for contributions from this factor, we were unable to locate the source regions of this
factor through air mass back trajectory frequency analysis (as described above). Therefore, local wind direction and normalized
contributions for this factor were used to create a conditional probability function (see Sect. 2.3). During the spring and autumn,
the dominant wind direction at Villum is from the southwest, while during the summer it is from the east (Nguyen et al., 2016).
The CPF can give information regarding the directional dependence of a factor or compound. Figure 8 shows the CPF for the
Background factor. There is a lack of directional dependence for this factor, indicating this factor does not arise from one
specific source area, but rather it is spatially ubiquitous.



The Background Factor likely represents natural processes occurring in the Arctic. This factor can serve as a baseline for comparison with future VOC measurements and source apportionment analysis. These comparisons can help expound upon the effects of climate change on the natural processes occurring in this pristine and sensitive region. This, however, requires more long-term VOC measurements, especially across all seasons.

### 3.3.4 Arctic Haze Factor

The Arctic Haze factor exhibits high contributions at the beginning of April and it rapidly decreases until the middle of May where it remains low and stable for the remaining of the measurement campaign (Fig. 9, top). This factor accounts for 56 % of the variation of benzene and zero percent of acetonitrile, which suggests fossil fuel combustion processes as the source of this factor (Liu et al., 2008) (Fig. 9, bottom). Interestingly, the other species apportioned to this factor with significant contributions, i.e., MEK, formic acid, formaldehyde, and $C_5H_8O^+$ (Fig. 9, bottom) are all oxygenated compounds that exhibit decreasing patterns in the spring as well as diurnal variation in the summer (Fig. 1 a, c, d, and f). Much like for the Background factor, the source of these OVOCs is the oxidation of liable organic carbon transported from the mid-latitudes.

The high levels of anthropogenic pollutants transported to the High Arctic during this period give the well-known 'Arctic Haze' phenomenon (Barrie et al., 1981). The decrease in mixing ratio during the spring is characteristic of the seasonality for long-range transport for this region (Willis et al., 2018). The mixing ratio of compounds emitted from sources outside the Polar dome is drastically reduced in the summer (Klonecki et al., 2003). Also, the faster oxidation rates due to higher OH radical concentrations as well as increased wet scavenging during transport in summer will reduce VOC and BC mixing ratios (Browse et al., 2012). Gautrois et al. (2003) reported benzene mixing ratios for 7 years at Alert, CA, and found an annual variation similar to observations for the Arctic Haze Factor in this study. The enhanced levels of BC (not shown) during this period (and lack thereof during summer and autumn) supports the assignment of this factor to anthropogenic combustion sources.

The Arctic Haze factor presented in this study can be compared to other Arctic Haze factors previously found using factor analysis or clustering of either aerosol composition or PNSD data. Lange et al. (2018) used k-means clustering of aerosol size distribution to classify the accumulation mode aerosol population from Villum. The authors found three accumulation mode clusters, one of which they named 'Haze' occurred predominantly in the winter/spring and was largely absent in the summer. The Haze cluster contained the largest amounts of refractory BC, sulfate, and organics as well as the highest concentrations of CCN. Extending this analysis into the chemical composition of aerosols, Nielsen et al. (2019) utilized PMF to find three factors. The factor deemed 'Arctic Haze Organic Aerosol' was closely correlated with sulfate and temporally followed the pattern exhibited by the Haze cluster from Lange et al. (2018) and the Arctic Haze Factor (this study), due to the contraction of the Polar Dome in spring. These similar factors/clusters resolved from different data sources (PNSD, aerosol chemical composition, and VOCs) and different statistical methods (k-means and PMF) highlight the extent of how anthropogenic pollution can influence the characteristics of the High Arctic atmosphere. Given recent trends in emission reductions across Europe and Eurasia, these factors/clusters are expected to decrease in magnitude, although the extent and



occurrence of this anthropogenic pollution will ultimately be governed by several factors including transport patterns, precipitation patterns, and expansion of anthropogenic pollution sources within the Arctic circle (resource extraction and shipping) (Law et al., 2017).

## 4 Conclusions

VOCs mixing ratios were measured during April–October 2018 at the High Arctic station Villum Research Station, located at
Station Nord in Northeast Greenland. We identified 10 compounds by PTR-ToF-MS and provided time series of VOCs in the High Arctic covering several months. Generally, the mixing ratios observed in the present study are in accordance with other VOC measurements carried out in Arctic locations. We apportioned sources of these VOCs using PMF, finding four factors: Biomass Burning, Marine Cryosphere, Background, and Arctic Haze. The Biomass Burning Factor exhibited maxima during the autumn and the chemical profile was dominated by acetonitrile with contributions from benzene. Back trajectory analysis
reveals the influence of fires in North America and Eurasia. Interestingly, BC did not show enhancements during the peak of the Biomass Burning Factor, which we show is due to washout during transport. The Marine Cryosphere Factor was described by carboxylic acids (formic, acetic, and propionic acid) and DMS. This factor displayed maxima in the summer during periods of snow and sea ice melt. Back trajectory analysis yielded MIZs around the coasts of Greenland and the Arctic Ocean as source regions. The Background Factor showed maxima in the spring, autumn, and minima during the summer. While acetone was
the dominating species in this factor, the chemical profile did not resemble any known processes or sources. Oxidation of liable organic carbon is proposed as the source of the OVOCs present in this factor. The Arctic Haze Factor peaked in April, decreased until mid-May, and was absent during the summer. This factor was driven by levels of benzene as well as OVOCs. The source of OVOCs present in this factor is postulated to be oxidation of precursor emissions during transport from the mid-latitudes to the Arctic.

This study has several important results that have implications for the Arctic climate. Recent studies have highlighted the importance of natural emissions to aerosol formation and their contribution to CCN concentrations in the summer (Leaitch et al., 2016; Lange et al., 2019; Nielsen et al., 2019). The Marine Cryosphere Factor presents an important source of condensable vapors necessary for this formation and growth to CCN sizes. Due to increasing temperatures in the Arctic, the snowpack and sea ice are expected to experience increased melting in the coming years, which could increase the flux of DMS
and carboxylic acids from the surface to the atmosphere. With the onset of the melt season in the Arctic expected to begin earlier in the future, we also expect that the timing of this onset can also affect NPF events and their subsequent growth as well as ozone photochemistry. While biomass burning is expected to increase in the future, the year-to-year variability is still highly uncertain. The Biomass Burning Factor was characterized by acetonitrile, benzene, and correlated temporally with ozone. Due to washout during transport, there were no enhancements in BC during the peak of the Biomass Burning Factor. The inter-
annual variability of biomass burning events and meteorological conditions can, therefore, have a substantial impact on atmospheric pollution levels at ground level.



While this research provides valuable insight into the atmospheric chemistry and sources of VOCs in the High Arctic, future work is still needed. While calculated mixing ratios using a kinetic quantification are reliable, they are inherently uncertain, therefore external calibration with gas-phase standards would greatly improve the accuracy and reduce the analytical uncertainty. This work presents a long time series of VOC mixing ratios; however, these measurements are only during polar day. A full seasonal cycle including polar night, dark to light transition periods, and polar day would help elucidate the importance of transport of anthropogenic emissions in the absence of photochemical reactions. This work expounds on the understanding of the atmospheric chemistry and sources of VOCs in the High Arctic; however, future research is needed to fully understand the biogeochemical feedback mechanisms and their implications for a changing Arctic.

*Data availability.* All data used in this publication are available to the community and can be accessed by request to the corresponding authors Jakob Boyd Pernov (jbp@envs.au.dk) and Rossana Bossi (rbo@envs.au.dk).

*Author contributions.* JBP, RB, and RH collected the measurements. JBP and RB processed the data. JBP, JH, RB, and TL analyzed the data. JBP and TL performed the PMF analysis. JBP and JH wrote the manuscript. All co-authors proofread and commented on the manuscript.

*Competing interests.* The authors declare that they have no conflict of interest.

*Acknowledgments.* Villum Foundation is gratefully acknowledged for financing the establishment of Villum Research Station and the instrumentation used in this study (PTR-ToF-MS). Thanks to the Royal Danish Air Force and the Arctic Command for providing logistic support to the project. Christel Christoffersen, Bjarne Jensen, and Keld Mortensen are gratefully acknowledged for their technical support. We acknowledge the use of data from the NASA FIRMS application (https://firms.modaps.eosdis.nasa.gov/) operated by the NASA/Goddard Space Flight Center Earth Science Data and Information System (ESDIS) project and NASA's Earth Science Data and Information System (ESDIS) with funding provided by NASA Headquarters. We acknowledge Chad Greene for use of MATLAB functions (Greene et al., 2017). We acknowledge Francesco Canonaco for helpful discussions on the PMF analysis as well as Ksenia Tabakova and Paul Glantz for help with HYSPLIT and MATLAB.

*Financial support.* This research has been financially supported by the Danish Environmental Protection Agency and the Danish Energy Agency with means from MIKA/DANCEA funds for environmental support to the Arctic region (project nos. Danish EPA: MST-113-00-140; Ministry of Climate, Energy, and Utilities: 2018-3767) and ERA-PLANET (The European Network for observing our changing Planet) Projects; iGOSP and iCUPE,.and finally by the Graduate School of Science and Technology, Aarhus University.





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





**Table 1.** Overview of measured protonated masses included in PMF analysis including mass to charge ratio of measured
protonated mass, empirical formula, assigned compound name, mean volume mixing ratio in ppbv, mean LOD in ppbv,
percentage below LOD, and mean relative uncertainty.

| Measured mass ($m/z$) | Empirical Formula | Assigned Compound | Mean (ppbv) | Mean LOD (ppbv) | % < LOD | Mean Relative Uncertainty (%) |
|---|---|---|---|---|---|---|
| 30.997 | $CH_2OH^+$ | Formaldehyde | 0.220 | 0.176 | 0.6 | 41 |
| 42.019 | $C_2H_3NH^+$ | Acetonitrile | 0.067 | 0.045 | 0 | 46 |
| 47.011 | $CH_2O_2H^+$ | Formic Acid | 0.454 | 0.250 | 17 | 37 |
| 59.062 | $C_3H_6OH^+$ | Acetone | 0.608 | 0.037 | 0 | 32 |
| 61.047 | $C_2H_4O_2H^+$ | Acetic Acid | 0.201 | 0.096 | 5 | 39 |
| 63.034 | $C_2H_6SH^+$ | Dimethyl Sulfide | 0.046 | 0.043 | 4 | 57 |
| 73.068 | $C_4H_8OH^+$ | Methyl Ethyl Ketone | 0.031 | 0.023 | 0.1 | 56 |
| 75.058 | $C_3H_6O_2H^+$ | Propionic Acid / Methyl Acetate | 0.025 | 0.031 | 0.1 | 61 |
| 79.057 | $C_6H_6H^+$ | Benzene | 0.027 | 0.031 | 0.5 | 64 |
| 85.066 | $C_5H_8OH^+$ | N/A | 0.027 | 0.030 | 0.03 | 61 |





**Table 2.** Input species for PMF model along with species categorization, S/N, and $R^2$ value for modeled versus measured values.

| Species | Categorization | S/N | $R^2$ (Modelled vs Measured) |
|---|---|---|---|
| Formaldehyde | Weak | 0.9 | 0.83 |
| Acetonitrile | Strong | 1.1 | 0.97 |
| Formic Acid | Weak | 1.0 | 0.67 |
| Acetone | Strong | 2.2 | 1.00 |
| Acetic Acid | Strong | 1.0 | 0.67 |
| Dimethyl Sulfide | Weak | 0.4 | 0.62 |
| Methyl Ethyl Ketone | Weak | 0.5 | 0.95 |
| Propionic Acid | Weak | 0.2 | 0.91 |
| $C_5H_8O$ | Weak | 0.2 | 0.62 |
| Benzene | Strong | 0.3 | 0.96 |




**Fig. 1.** Time series of mixing ratios (ppbv) for **(a)** formaldehyde, **(b)** acetonitrile, **(c)** formic acid and acetic acid, **(d)** acetone and MEK, **(e)** DMS and propionic acid, and **(f)** benzene and $C_5H_8O$ during the entire measurement period.






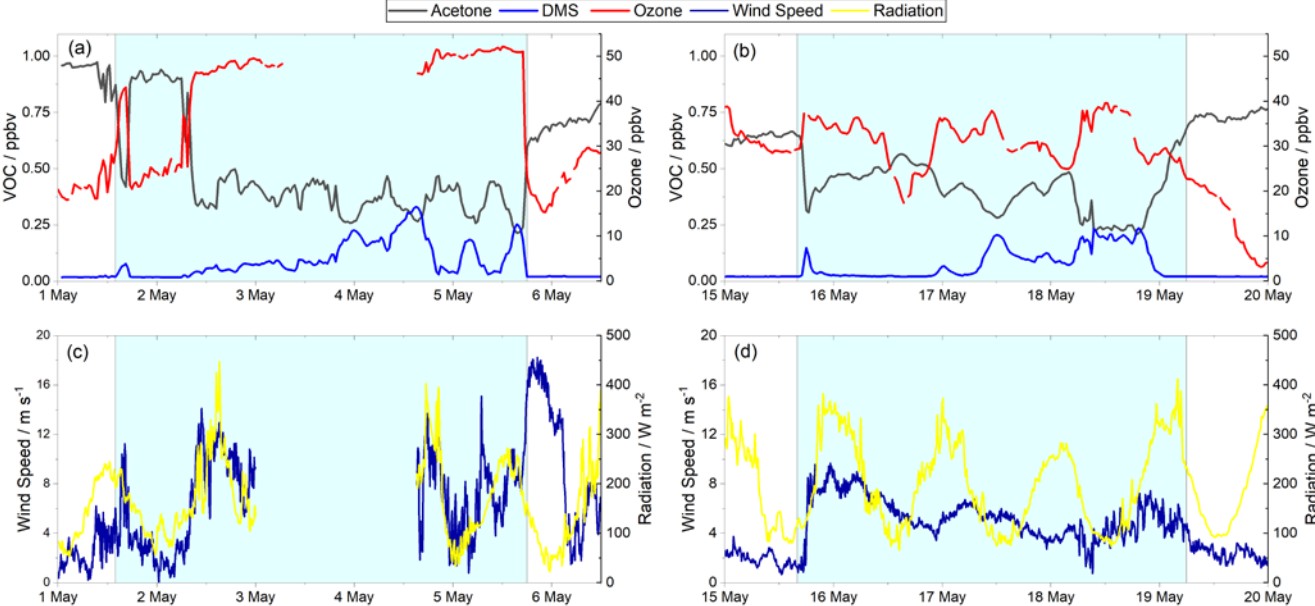

**Fig. 2.** Left: The first period of elevated DMS mixing ratios (May 1–5). Right: The second period of elevated DMS mixing ratios (May 15– 19); (a) and (b) mixing ratios of acetone, DMS (left axis), and ozone (right axis); (c) and (d) radiation (left axis) and wind speed (right axis). The shaded area represents episodes of elevated DMS mixing ratios.




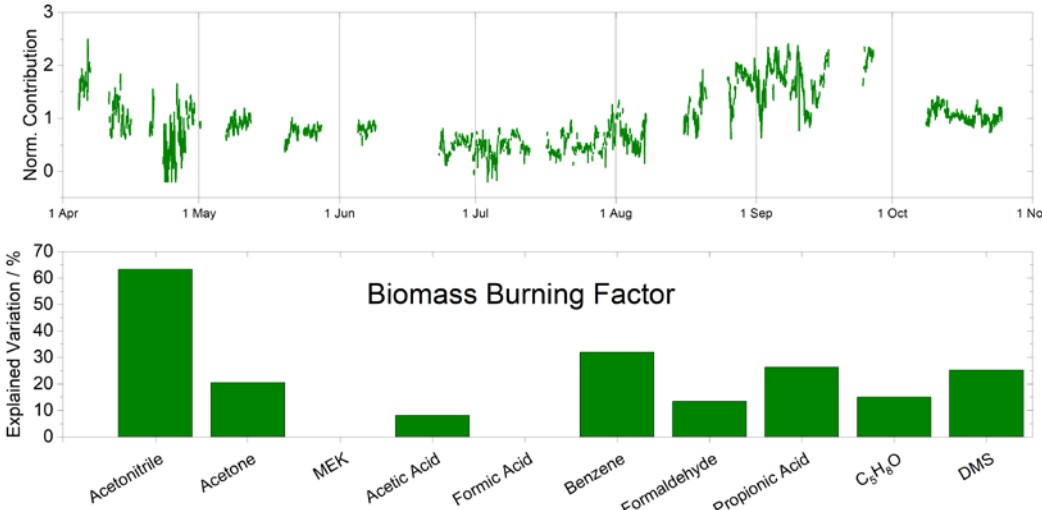

**Fig. 3. (Top)** Time series of normalized contributions and **(Bottom)** species profile for the Biomass Burning factor. Factor contributions are normalized to give a mean contribution of unity.




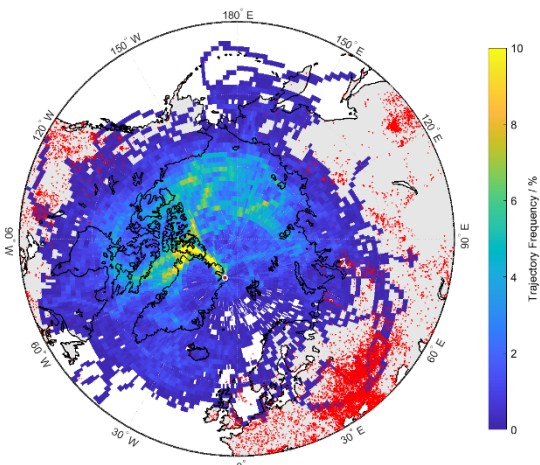

**Fig. 4.** HYSPLIT air mass back trajectory frequency maps arriving at 100 m altitude extending 336 hours backward in time. Active fire data from FIRMS are shown in red stars. The location of Villum is shown as a red and white circle. The period only includes the peak of the Biomass Burning factor (August 15–September 15, 2018).






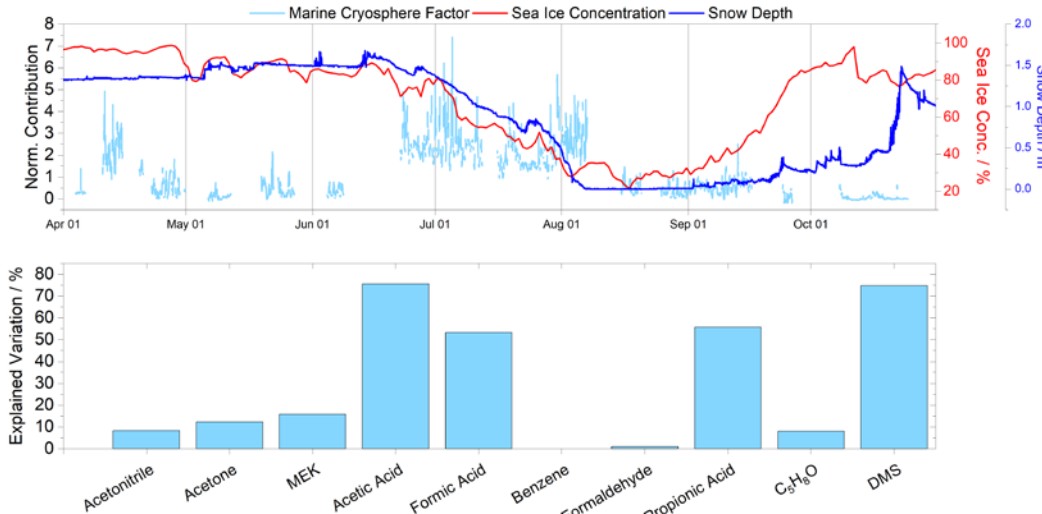

**Fig. 5 (Top)** Time series of normalized contributions in light blue, left axis, and sea ice concentrations in red and snow depth in blue, right axes, and **(Bottom)** species profile for the Marine Cryosphere factor. Factor contributions are normalized to give a mean contribution of unity.



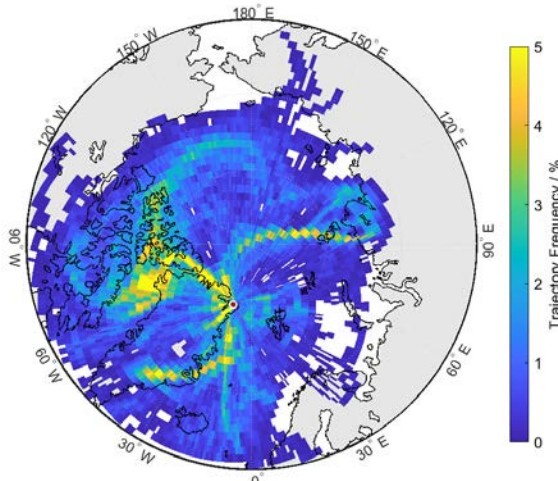

**Fig. 6.** Trajectory frequency map for air mass back trajectories arriving at 100 m altitude, extending backward 240 hours in time. Trajectory frequency calculation is described in Sect. 2.4. The color bar is capped at 5 % for visual clarity.






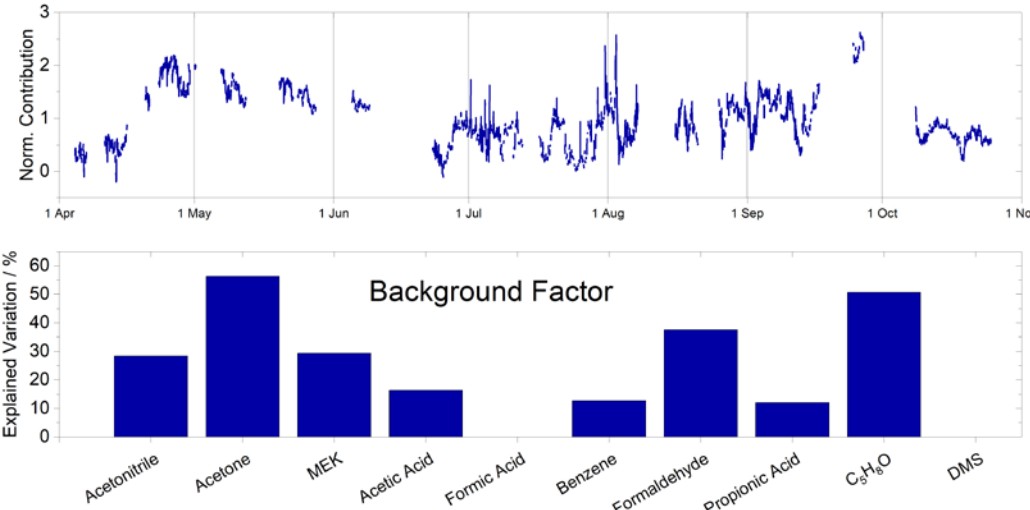

**Fig. 7. (Top)** Time series of normalized contributions and **(Bottom)** species profile for the Background factor. Factor contributions are normalized to give a mean contribution of unity.





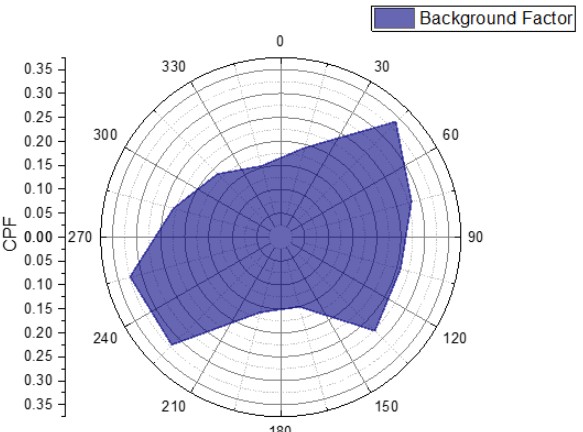

**Fig. 8.** Conditional Probability Function for the Background Factor. CPF was calculated as described in Sect. 2.3.

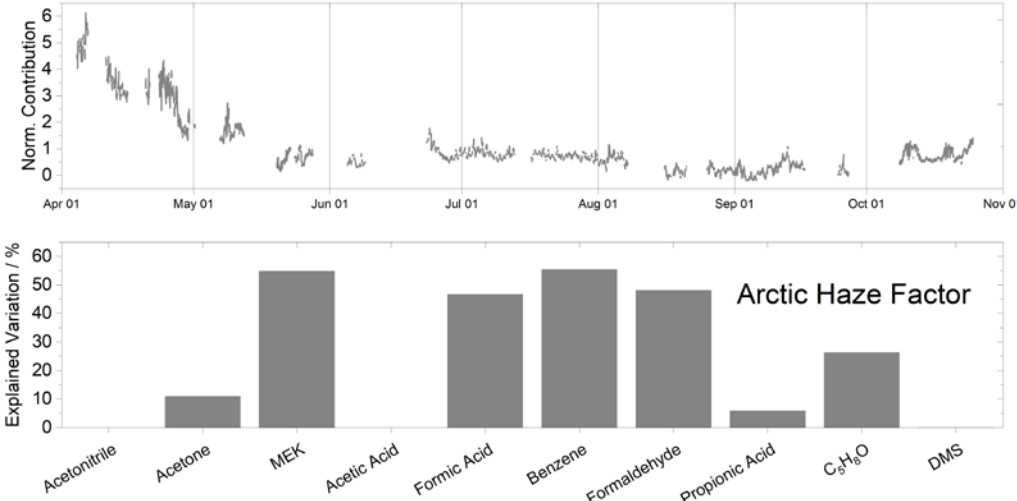

**Fig. 9. (Top)** Time series of normalized contributions and **(Bottom)** species profile for the Arctic Haze factor. Factor contributions are normalized to give a mean contribution of unity.
