# Peer review of "1. Mixing Ratio and Uncertainty Calculation"

_Atmospheric Chemistry and Physics, 2020_

## Referee Comment (RC1) · Anonymous Referee #1 · 25 Aug 2020

This manuscript reports on a long-term (spring through fall) Arctic VOC dataset observed at Villum Research Station at Station Nord in Greenland, and a PMF analysis performed on the dataset. The authors report the PTR-ToF-MS results for 10 detected ions, assigning 10 gas-phase molecular formulae and species/species groups to the observed ions in the PTR. A PMF analysis of the 10 species and species groups with a four-factor analysis is presented and discussed at length, including a Biomass Burning Factor, a combination Marine Cryosphere Factor, a Background Factor, and an Arctic Haze Factor. The authors give a very nice detailed analysis of the four factors, including the primary components, sources and influences and temporal variability.

[Figure]

Overall this is a well-written paper, with valuable data and analysis that should be published. One of my primary concerns with the paper, and with the majority of PTR-instrument papers, is that there is a lack of accounting or explanation of the VOC specificity (or lack thereof) of the PTR technique. The authors make no effort in this paper to discuss the interfering or additional species that may comprise each observed chemical formula that make up several of their measurements – e.g., propanal's contribution to the signal attributed to acetone, isobutanal's and butanal's contribution to the signal attributed to MEK – even to justify the omission of these species from the discussion with adequate explanation and literature references. As well, the authors' treatment of methyl acetate and propionic acid is to suggest that the contributions from each species (or other species that might contribute to the C3H6O2H+ signal) are unknown in Section 2.2, but then they attribute the signal to methyl acetate in the Biomass Burning Factor, and propionic acid in the Marine Cryosphere Factor, with no justification as to the reasons for the identification. The authors need to add commentary for the species identification, and justify the assumed VOCs under different conditions, or simply refer to the observations as a generic C3H6O2 VOC group. Also, as detailed below, references to VOCs that comprise the C5H8O observation should be clear that the measurement is not of an ion (C5H8OH+ or C5H8O+), but of the C5H8O VOC group.

My other primary concern is that the authors indicate that the data are available by contacting one of two author email addresses. I would strongly recommend that the paper not be published until the data are available in a publicly-available DOI.

The remainder of my comments are minor and detailed below.

As stated at the end of Sect. 3.2, "The different temporal patterns and correlations suggest the behavior and sources of VOCs in the Arctic are seasonally dependent. Therefore, a detailed, statistical investigation of the sources affecting VOC levels is warranted." This is very true, and the reason why this paper should be published.
Lines 145-152 – the authors describe the method by which "compound names" are assigned to the nine protonated masses, including Pagonis et al. and references, which is reasonable, and a priori knowledge, which is not something that can be reference checked. I would argue that there is insufficient justification given to identifying the masses which ignore contributions from additional compounds that may be included in the concentrations measured. The authors write "Another compound (C4H8OH+) was doubly assigned to propionic acid and methyl acetate.", but they likely meant to write C3H6O2H+, which has m/z 75.058. However, they should explain here why they don't include ethyl formate or hydroxyacetone as possible compounds at this mass.

Lines 155-157 – The authors should be specific about how the data were quality controlled using these parameters (PSND, WD, WS, etc.), and the resulting amount (total percentage, number of time periods, etc.) of data that had to be eliminated from the useful data set.

Lines 215, 212, 467, 506, etc. – Technically the authors did not observe ambient C5H8O+ ions (or C5H8OH+ ions), but rather a compilation of [some] gas-phase C5H8O species, which were protonated in order to be observed by the PTR system, similar to how they did not observe atmospheric ambient C6H6H+ ions, but rather gas-phase C6H6 (i.e., benzene). Thus, discussion of the species or group of species with the chemical formula C5H8O should simply be "C5H8O species" or "C5H8O", as in Figure 1, and should not imply the measurement of an atmospheric ion.

Lines 245-252 – The comparisons presented against literature data from similar Arctic stations make sense, for the most part, but the comparison of wintertime benzene mixing ratios from Gautrois et al. (2003) to this study are not merited, as no wintertime data is being presented here. As well, while I agree that it has been shown that benzene and acetonitrile are influenced by lower latitudes, the claim that acetonitrile is influenced by anthropogenic emissions is not backed up. Remote levels of acetonitrile are likely impacted by the significance of mid-latitude fire seasons, and are not expected to compare well from year to year.

Line 308 – the authors state that species with S/N < 0.2 were excluded from the analysis, but all 10 species (or species groups) discussed in the paper are included in Table 2. Are there any other species that were measured but not included here?

Line 337 – "it is a source of methyl acetate as well..." – the authors recognize that methyl acetate could be contributing to the C3H6O2H+ signal, but by labeling it "propionic acid" in Table 2 and Figures 1, 3, 5, 7, etc., the identity of the compound is muddied. If the authors truly believe that the species is primarily propionic acid, then the presence of methyl acetate would be unimportant. If they believe that it is indeed a mixture of the two (or more) species, then this should be made clear whenever it is being referred to.

Line 445 – The back trajectories frequency map for the Marine Cryosphere Factor is interesting, but it would be more informative to highlight some of the brief periods where this factor is particularly elevated, rather than averaging over a three-month summer period. Given, as well, that all the species identified to contribute to the Marine Cryosphere Factor have atmospheric lifetimes < 5 days, it would be prudent to limit these back trajectories to 120 hours or less.

Figures – all figures in the primary manuscript and supplement should be saved at a higher resolution. There is significant pixilation when zooming in on the plots. Some of the finer details are lost as a result, and some of the axis labels are rendered illegible.

Table 1 – The table title doesn't need to be so long. "Overview of measured protonated masses included in PMF analysis" would be sufficient. The rest is redundant with the table header, although Mean Mixing Ratio should be spelled out in the header or defined in a footnote. As well, it would be good to specify if the "Percentage below LOD" is the percentage of all data collected, or the percentage of only the data that was not removed due to the influence of local pollution. The same comment goes for the means reported.

Technical corrections

Line 27 – "rate" would be preferable to "speed".

Line 33 – NOx should be defined.

Line 46 – there is a rogue hyphen/em dash that isn't needed.

Line 46 and others – Dall'Osto is missing an apostrophe both here and in the reference list, where the reference is also missing several other diacritical marks, and the majority of C.D. O'Dowd's last name. The references should then be rearranged for this reference to come before the more recent Dall'Osto et al. references. Be wary of automatic reference management software – references should still be verified that they were transposed and recorded properly.

Line 68 – "loss" rather than "reactions" would generate better agreement with the singular "sink".

Lines 90-93 – "Furthermore, Boudries et al. (2002) observed emission from the snowpack to the atmosphere of acetone, acetaldehyde, and formaldehyde, which were explained by photochemical production in the snowpack and depositional fluxes of methanol was also observed, which they postulated as a source of formaldehyde." - Consider making this two sentences: "Furthermore, Boudries et al. (2002) observed emission from the snowpack to the atmosphere of acetone, acetaldehyde, and formaldehyde, which were explained by photochemical production in the snowpack. "Depositional fluxes of methanol were also observed, which they postulated as a source of formaldehyde" Or at the very least, add a semi-colon and change "was" to "were" on line 92.

Line 94 – there should be a comma after "VOCs".

Line 103 – It would be good to mention that Barrow, AK is now Utqiagvik, AK.

Line 104 and others – While "Alert, CA" is technically acceptable, "Alert, Canada,", or "Alert, NU," would be less ambiguous. Also, be consistent throughout. Greenland should probably be spelled out as well.

Line 122 – Use "s" instead of "sec" to adhere to SI units requirement.

Line 122 – Use "southwest" instead of "south-west".

Line 131 – "5 seconds scan rate" doesn't describe a rate, which should be something per unit of time.

Line 154 – "mixing ratios below LOD were set to"

Line 155 – "the data were time-averaged to 30-minute means."

Line 194 – "arriving from"

Lines 199-200 – "Active fires during the period 15 August – 15 September 2018 were provided..." (you are defining the period here, not referring to it, so the commas are not needed.)

Line 259 – I recommend splitting this long sentence, "... frozen sea surface. Back trajectory..."

Line 267 – "strong negative correlation" is a little too generous for R = -0.68.

Lines 271-273 – It would be informative to include wind direction in addition to wind speed in Figure 2.

Line 275 – "with elevated acetone levels during ozone..." or something similar.

Line 279 – "gas-phase"

Line 303 – "species with S/N..."

Line 304 – "The uncertainties of 'Weak' species were tripled..."

Line 314 – "VOCs devoid of episodic influence...", and there is a period missing at the end of the sentence.

Line 394 – The authors write "Estimated globally averaged atmospheric lifetimes against wet deposition for formic and acetic acid in the boundary layer is between 1

and 2 days respectively (Paulot et al., 2011)." – This is not clear. Are both of the estimated atmospheric lifetimes between 1 and 2 days? If so, "respectively" isn't needed. Either way, though, it should state "are between"...

Line 396 – "14C" (with a superscripted 14) or "carbon-14" (without a superscript).

Lines 405, 407, 410, 412, 443, 446, Figure 5, etc. etc. – sometimes "Factor" is capitalized in reference to one of the four factors, and sometimes it isn't. This should be consistent throughout.

Line 427 – "Factor", not "Factors". Also, there is an extra period in this sentence: "... speed (Fig. S2.)."

Lines 430-431 – Despite what the papers might claim, MSA is not measured in particle phase, but rather they measured the methanesulfonate ion, $CH_3SO_2^+$. It would be better to simply indicate that the presence of gas-phase MSA has been indicated by the observation of methanesulfonate ion in particles.

Line 438 – "... Dibb and Arsenault (2002) measured levels..."

Line 440 – "matter, e.g.,"

Line 444 – The sentence "These trajectories and trajectory frequency maps were calculated as described in Sect. 2.4." isn't necessary.

Lines 460-461 – recommend: "One of the source areas identified in Fig. 6 is southeast of Villum, and a CPF analysis indicated high contributions (of what?) were observed when the winds were from south of Villum (Fig. S8a)." – this sentence needs a little clean-up for readability and clarity.

Line 469 – Recommend changing "Most of its components, particularly acetone and formaldehyde, are known..." to simply "Acetone and formaldehyde are known..."

Lines 483, 484, 487, 508, 545 – do you mean "labile [organic] carbon"?

Line 531 – Circle should be capitalized.

References – Please format all references properly: pay attention to things like consistent journal abbreviations, consistent DOI referencing, missing or n/a information (e.g., line 680), line wrapping (e.g., line 735), and capitalization of abbreviations and proper names (e.g., lines 632; 839; 842, etc.).

Figure 4 – "red stars" – the resolution doesn't merit calling these stars. They're mostly just dots.

Supplement

Line 26 – Either "(5 s)" or "(5 seconds)" would be acceptable SI units.

Table S1 - The way the authors divided up the seasons here seems oddly arbitrary. Why is "summer" only two months long, while autumn is three months? And changing seasons on the 7th of a month is oddly arbitrary. As well, it would be preferable to separate the measurement and units in the first column with a comma rather than a slash. Also, use either "autumn" or "fall" but not both in the table title and header. Lastly, the start and stop dates in the title are not consistent with the dates given on Line 128 of the main text. Please make these consistent.

Tables S2-S4 – It is unclear why June, July and September are included here, but not August and October. In the text, Villum Research Station is referred to as "Villum", not VRS. It should be the same here, or spelled out in full. The vertical alignment of these tables is off, with the numbers right justified, and the headers left-justified, making it difficult to know which values go with which headers. As well, some of the compounds listed in the left-hand column blend together. Either increase the spacing, or shorten the names (i.e., MEK, DMS, etc.) to limit the amount of word-wrapping. Formic Acid across the head is also rather unfortunately split. Finally, the "All correlations, apart from . . ." in the titles should just be included as a footnote.

Figure S1 – the text suggests that there were times when the wind speeds were < 2

m/s, but this is not included in the figure. Please either include these, or justify their omission. Also, the resolution on the figure does not allow for the reader to discern anything > 14-18 m/s (blue). Either improve the resolution, or change the legend to eliminate the highest wind speed categories.

Figure S2 – "Time series of meteorological parameters. . ."; consider adding wind direction to this figure as well.

Figure S4 (and S5) – there is a lot of information shown that is repetitive and unneeded to the right of each satellite image, and as a result the majority of the important details are illegible. Remove the unnecessary parts, and make higher res and/or larger versions of the plots, and label the leads and the station in the image(s). As well, the labels a-f should be moved to the top left, or top right, or could be included inside the images in white for clarity. Lastly, here and throughout the manuscript, re: the ACP style guide, dates should be in the form dd month yyyy (or simply dd month).

Figure S6 – caption "A new trajectory was [calculated/generated] every 24 hours." The back trajectory trace colors in the plots should have a legend or be described.

Figure S7 – The caption should include the fact that this is from the PMF analysis.

Figure S8 – plots (a) and (c) have the same size CPF scale, but different numbers of ticks and significant figures. They should be the same.

---

## Referee Comment (RC2) · Anonymous Referee #2 · 20 Oct 2020

This manuscript by Pernov et al. reports atmospheric non-methane volatile organic compounds (VOCs) measurements at Villum Research Station at Station Nord, Greenland, from April to October 2018. Given the scarcity of VOC measurements in the Arctic and the significance of VOCs in the background atmosphere (formation of ozone, CO, and aerosols), this study will make a valuable contribution to the body of literature. The manuscript is overall well written and structured. My main concern is that the figures do not support the discussion and conclusions (see comments below). Additionally, the introduction could be more succinct.

- Diurnal variation

The authors say that certain compounds (e.g. DMS and OVOCs) follow a diurnal cycle. This is not shown in Figure 1 and I would like to see a Figure describing, for each compound of interest, the mean diurnal cycle per season.

- Springtime DMS

I have difficulties reading satellite images (Figures S4 and S5). The caption says that the presence of open leads can be seen southwest of the station but I don't even know where the station is. Then, according to the authors, "the back-trajectory calculations confirmed that, during the DMS emission episodes, the air masses (. . .) traversed over the open leads before reaching the station". First of all, what is the meaning of the different colors? I do not understand Figure S6. Then, this Figure does not show that air masses traversed over the open leads. If you want to show this, then please consider combining satellite images and back-trajectories on a single Figure.

- Biomass burning

Contrary to what the authors say, Figure 4 does not show "evidence of overlap between air mass history and active fires during this period". Figure 4 shows all fires from mid-August to mid-September and all back-trajectories. This does not prove that a given fire existed at the time an air mass traveled in the area. I expect a more thorough statistical analysis here. In order to link the fires up with back trajectories, you could for instance cross check the latitudes and longitudes to a, let's say, 1-degree accuracy. If a longitude and latitude match exists between a fire and a back-trajectory, then check if the time of the fire product and the back-trajectory were within, let's say, 1 hour. Thus, a match is completely defined as a back-trajectory crossing over a fire within 1 hour and within 1-degree difference.

- Spatial origin of the Marine Cryosphere factor

Figure 6 (trajectory frequency) shows more frequent air masses from coastal regions but does not show that these areas are responsible for enhanced marine cryosphere

factor. It does not support this sentence in the conclusion "Back trajectory analysis yielded MIZs around the coasts of Greenland and the Arctic Ocean as source regions". I suggest a Potential Source Contribution Function (PSCF) analysis to determine probable locations of emission sources.

Line-by-line comments:

Line 10: "we report a long-term dataset". The authors report measurement from April to October 2018, i.e., less than a year. This is not what I would call a "long-term dataset". Please edit this sentence.

Line 33: Define VOCs and NOx.

Line 49: Define DMS.

Line 108-109: "with low time resolution". Be more specific here. Gautrois et al. (2003) collected about one sample every 9 days. Additionally, the authors did not use a GC-MS, but a combination of GC-FID and GC-ECD.

Line 110: You don't really explain why we need high time-resolved measurements of VOCs. Do you expect a high temporal variability? How about the global atmospheric watch reactive gases measurement network (Schultz et al., 2015) – Aren't these measurements enough?

Line 111: Same comment as above. In the previous sentence, you highlight the need for long-term measurements of VOCs in the Arctic. While a substantial contribution to the literature, you "only" report several months of data. You could perhaps emphasize more the high temporal frequency of your measurements.

Line 123: Did you filter data for local contamination? If so, how?

Line 125, Table S1: I have a hard time understanding how the seasons were defined. Skov et al. (2020) recently used a different (and more straightforward) definition: winter from December to February, spring from March to May, summer from June to August,

and fall from September to November.

Line 137-138: "measurements were interrupted for short periods ranging from days to weeks". Could you please add a Table summarizing, for each month, the number of hours of operation?

Line 144: "within the analytical uncertainties". Please refer to Table 1 here. Additionally, how often did you perform a calibration?

Line 157: How exactly did you remove the influence of local pollution. What criteria did you use for wind speed and direction?

Lines 179-181: Did you perform a sensitivity test? How does changing concentrations below LOD and missing concentrations influence the PMF results?

191: "automatic weather station placed close to the measurement site". How close? Be more specific.

Lines 206-207: "The trajectory length was varied between 240 and 336 hours". Why did you use two different trajectory lengths? Additionally, I would recommend the use of shorter back trajectories (typically 5-7 days max) as uncertainties increase with time along the way (Stohl, 1998). I would also like to see a more critical discussion on back trajectories; they only give a general indication of source regions.

Line 217: "certain compounds (DMS and OVOCs) revealed a diurnal cycle that closely follows radiation". Please make a Figure to prove this.

Line 219: "summer when a diurnal pattern following sunlight was observed". Same as above, please demonstrate this.

Lines 225-227: "a clear diurnal variation was observed in the period July-August, with peak mixing ratios occurring around midday (Fig. 1 a, c, d, e). The diurnal variation was less pronounced in April-May and September-October, highlighting the dependence on sunlight". None of this is shown in Figure 1.

Line 255: "DMS showed a clear diurnal cycle during sea ice melt in the summer months correlating with sunlight intensity". Prove/illustrate it.

Line 257: "Elevated DMS mixing ratios". What do you mean by "elevated"? Be more specific.

Lines 259-261: see comment above. As is, Figures S4-S6 do not do a good job at showing this.

Line 273: "illustrated here by changes in wind speed". I would expect changes in wind direction to be a more useful tracer of change in meteorological conditions.

Lines 286-287: "In addition to the previously mentioned dependence on the diurnal variations of sunlight, providing strong evidence of a local photochemical source". Again, this has not been demonstrated.

Line 344: "a sink during the summer". A sink of what?

Line 351: "336 hours backward in time". That's too long to my point of view. Use max 5-7 days.

Lines 383-384: "This factor shows an enhanced diurnal variation with a clear correlation to sunlight during the summer months (Fig. 5, Top)". Again, Figure 5 does not illustrate this.

Lines 406-407: "Periods of high contributions and clear diurnal pattern by the Marine Cryosphere factors starts on June 23". I don't see the "clear diurnal pattern".

Lines 426-427: "Although, the variation of the Marine Cryosphere Factors seems not to be driven mainly by the dependence on horizontal wind speed (Fig. S2)". Figure S2 does not illustrate this. What is the correlation coefficient between the Marine Cryosphere Factor and wind speed?

Lines 428-429: "given the distance of the measuring site from open water". What is the distance between the station and open water?

Lines 443-454: see comment above on Figure 6 and the fact that it does not show that coastal regions are responsible for enhanced marine cryosphere factor.

Section on Arctic Haze: please mention/discuss more clearly that you do not have data in wintertime, when Arctic Haze is expected to be at its maximum.

References

Gautrois, M., Brauers, T., Koppmann, R., Rohrer, F., Stein, O. and Rudolph, J.: Seasonal variability and trends of volatile organic compounds in the lower polar troposphere, , doi:10.1029/2002JD002765, 2003.

Schultz, M. G., Akimoto, H., Bottenheim, J., Buchmann, B., Galbally, I. E., Gilge, S., Helmig, D., Koide, H., Lewis, A. C., Novelli, P. C., Dülmer, C. P.-, Ryerson, T. B., Steinbacher, M., Steinbrecher, R., Tarasova, O., Tørseth, K., Thouret, V. and Zellweger, C.: The Global Atmosphere Watch reactive gases measurement network, Elem Sci Anth, 3(0), 000067, doi:10.12952/journal.elementa.000067, 2015.

Skov, H., Hjorth, J., Nordstrøm, C., Jensen, B., Christoffersen, C., Bech Poulsen, M., Baldtzer Liisberg, J., Beddows, D., Dall'Osto, M. and Christensen, J.: The variability in Gaseous Elemental Mercury at Villum Research Station, Station Nord in North Greenland from 1999 to 2017, Atmospheric Chemistry and Physics Discussions, 1–22, doi:https://doi.org/10.5194/acp-2019-912, 2020.

Stohl, A.: Computation, accuracy and application of trajectories - a review and bibliography, Atmospheric Environment, 32(6), 947–966, 1998.

---

## Author Comment (AC2) · 2 Dec 2020

Reply to *Interactive Comment* on "Atmospheric VOC measurements at a High Arctic site: characteristics and source apportionment" from anonymous Referee # 2

This manuscript by Pernov et al. reports atmospheric non-methane volatile organic compounds (VOCs) measurements at Villum Research Station at Station Nord, Green- land, from April to October 2018. Given the scarcity of VOC measurements in the Arctic and the significance of VOCs in the background atmosphere (formation of ozone, CO, and aerosols), this study will make a valuable contribution to the body of literature. The manuscript is overall well written and structured. My main concern is that the figures do not support the discussion and conclusions (see comments below). Additionally, the introduction could be more succinct.

We would like to thank referee # 1 for carefully reading the manuscript and for useful comments and feedback. We feel it improved the manuscript's readability and overall discussion of the results. As the first author is an early career scientist, they feel this exercise in the peer-review has tremendously helped them progress in critical thinking, manuscript writing, and the scientific method. We have addressed the referee's concerns and corrected errors in the manuscript below with referee's comments numbered and the author's responses in blue. New references are highlighted in yellow.

Several of the reviewer's comment address the same concerns, where appropriate we have grouped these comments together and responded to them all with one reply.

The referee suggested the Introduction could be more succinct. We have removed the following lines from the introduction in order to reduce the wordiness:

Lines 40–44

Lines 64–65

Lines 73–75

1)  Diurnal variation

The authors say that certain compounds (e.g. DMS and OVOCs) follow a diurnal cycle. This is not shown in Figure 1 and I would like to see a Figure describing, for each compound of interest, the mean diurnal cycle per season.

Line 217: "certain compounds (DMS and OVOCs) revealed a diurnal cycle that closely follows radiation". Please make a Figure to prove this.

Line 219: "summer when a diurnal pattern following sunlight was observed". Same as above, please demonstrate this.

Lines 225-227: "a clear diurnal variation was observed in the period July-August, with peak mixing ratios occurring around midday (Fig. 1 a, c, d, e). The diurnal variation was less pronounced in April-May and September-October, highlighting the dependence on sunlight". None of this is shown in Figure 1.

Line 255: "DMS showed a clear diurnal cycle during sea ice melt in the summer months correlating with sunlight intensity". Prove/illustrate it.

Lines 259-261: see comment above. As is, Figures S4-S6 do not do a good job at showing this.

Lines 286-287: "In addition to the previously mentioned dependence on the diurnal variations
of sunlight, providing strong evidence of a local photochemical source". Again, this has not
been demonstrated.

Lines 383-384: "This factor shows an enhanced diurnal variation with a clear correlation to
sunlight during the summer months (Fig. 5, Top)". Again, Figure 5 does not illustrate this.

Lines 406-407: "Periods of high contributions and clear diurnal pattern by the Marine
Cryosphere factors starts on June 23". I don't see the "clear diurnal pattern".

We agree with the referee's concern about a lack of the diurnal nature being illustrated properly.
We have therefore added two figures in the main text (Fig. 2 and 6) to display diurnal profile
for each of the relevant compounds and the four factors, respectively, during the summer
months, as well as the diurnal profiles for each season (as requested by the referee) in the
Supplement (Fig. S3, S4, and S5). We have amended the text throughout to reference these two
figures and removed references to a diurnal profile when they do not pertain to these figures.

[Figure]

**Fig. 2.** Diurnal profile for (a) formic acid, acetone, acetic acid, and radiation and (b) MEK,
formaldehyde, $C_3H_6O_2$, DMS, and radiation during the period 22 June–09 August.

[Figure]

**Fig. 6.** Time series of the four factors from 22 June–09 August displaying the diurnal profile
together with radiation.

**Diurnal Profile for Spring**

[Figure]

**Fig. S3.** Diurnal profile for the spring (April–May) of **(a)** formaldehyde, **(b)** acetonitrile, **(c)**
formic acid, **(d)** acetone, **(e)** acetic acid, **(f)** DMS, **(g)** MEK, **(h)** $C_3H_6O_2$, **(i)** benzene, **(j)** $C_5H_8O$.
Data were averaged to hourly medians.

**Diurnal Profile for Summer**

[Figure]

**Fig. S4.** Diurnal profile for the summer (June–August) of **(a)** formaldehyde, **(b)** acetonitrile,
**(c)** formic acid, **(d)** acetone, **(e)** acetic acid, **(f)** DMS, **(g)** MEK, **(h)** $C_3H_6O_2$, **(i)** benzene, **(j)**
$C_5H_8O$. Data were averaged to hourly medians.

**Diurnal Profile for Autumn**

[Figure]

**Fig. S5.** Diurnal profile for the autumn (September–October) of **(a)** formaldehyde, **(b)** acetonitrile, **(c)** formic acid, **(d)** acetone, **(e)** acetic acid, **(f)** DMS, **(g)** MEK, **(h)** $C_3H_6O_2$, **(i)** benzene, **(j)** $C_5H_8O$. Data were averaged to hourly medians.

We have amended the text to mention the diurnal profiles for relative compound in the relative season:

Line 269–280: "For the ten selected VOCs, time series of mixing ratios during the entire measurement period are displayed in Fig. 1a-f. During the spring (April–May), certain compounds (benzene and $C_5H_8O$) exhibited a maximum and thereafter a decreasing pattern, similar to the timing and profile of the Arctic Haze phenomena. During the spring, compounds did not display a diurnal profile except for acetic acid (Fig. S3) Whilst in summer (June–August), OVOCs revealed a diurnal cycle that closely follows radiation (Fig. 2 and S4). Compounds of non-photochemical origin (benzene and acetonitrile) also displayed a slight diurnal pattern, which could possibly be due to entrainment from aloft (Fig. S4). Interestingly, several compounds (formaldehyde, formic acid, and acetone) peaked in the spring with decreasing levels until the summer when a diurnal pattern following sunlight was observed (Fig. 1, 2, S4). During the autumn (September–October), all compounds were low except for acetone and acetonitrile (Fig. 1) and only acetic acid displayed a diurnal profile (Fig. S5). The levels, seasonal patterns, and comparison with other studies of these compounds will be discussed below."

2) Springtime DMS

I have difficulties reading satellite images (Figures S4 and S5). The caption says that the presence of open leads can be seen southwest of the station but I don't even know where the station is. Then, according to the authors, "the back-trajectory calculations confirmed that, during the DMS emission episodes, the air masses (. . .) traversed over the open leads before reaching the station". First of all, what is the meaning of the different colors? I do not understand Figure S6. Then, this Figure does not show that air masses traversed over the open leads. If you want to show this, then please consider combining satellite images and back-trajectories on a single Figure.

Figures S4 and S5 have been removed from the manuscript. Both reviewers raised concerns about the legibility of these two figures, therefore, we have removed them and directed the reader to the website where they were obtained (http://ocean.dmi.dk/arctic/nord.php). We feel they add valuable information about the origin of the elevated DMS periods but displaying them in a meaningful manner proved problematic. We have left the HYSPLIT back trajectories in Fig. S6 in and updated the figure caption to indicate the meaning of the different trajectories, the text now reads :

"HYSPLIT back trajectory analysis for **(a)** May $2^{nd}$– $6^{th}$ **(b)** May $16^{th}$–$20^{th}$ arriving at 100 m above ground level extending 72 hours backward in time. The colored trajectories represent a new trajectory started every 24 hours after the last day of each period until the first day, in descending order the trajectories are red (last day), blue (fourth day), green (third day), light blue (second day), and purple (first day)."

3) Biomass burning

Contrary to what the authors say, Figure 4 does not show "evidence of overlap between air mass history and active fires during this period". Figure 4 shows all fires from mid- August to mid-September and all back-trajectories. This does not prove that a given fire existed at the time an air mass traveled in the area. I expect a more thorough statistical analysis here. In order to link the fires up with back trajectories, you could for instance cross check the latitudes and longitudes to a, let's say, 1-degree accuracy. If a longitude and latitude match exists between a fire and a back-trajectory, then check if the time of the fire product and the back-trajectory were within, let's say, 1 hour. Thus, a match is completely defined as a back-trajectory crossing over a fire within 1 hour and within 1-degree difference.

Lines 206-207: "The trajectory length was varied between 240 and 336 hours". Why did you use two different trajectory lengths? Additionally, I would recommend the use of shorter back trajectories (typically 5-7 days max) as uncertainties increase with time along the way (Stohl, 1998). I would also like to see a more critical discussion on back trajectories; they only give a general indication of source regions.

Line 351: "336 hours backward in time". That's too long to my point of view. Use max 5-7 days.

We collocated back trajectory endpoints with active fires with 1° latitude/longitude and temporally within one hour as the reviewer requested. While there was evidence of active fires in North America and Eurasia occurring when an endpoint was near (see figure below), as the reviewer pointed out the uncertainty in individual trajectories at 336 hours is too great to draw meaningful conclusions from this analysis. Therefore, this figure has been removed from the manuscript.

We have included the figure in our response, this figure will not be included in the manuscript. Individual trajectories are indicated in the dashed blue lines and active fires occurring within 1° lat/lon and within one hour of trajectory endpoints are indicated in red.

[Figure]

We have amended the text for the Biomass Burning section.

Line 420–431: "To examine the geographical origin of this factor, air mass back trajectories
from the HYSPLIT model were calculated every hour during the peak of the Biomass Burning
Factor (15 August–15 September 2018) and extending 336 hours (two weeks) backward in
time. The trajectory length of two weeks was selected to account for the long lifetime of
acetonitrile. Active fires during the period 15 August–15 September 2018 were provided by
NASA's Fire Information for Resource Management System (FIRMS) (Schroeder et al., 2014).
Active fires occurring with one hour and one-degree latitude/longitude of a trajectory endpoint
was used to access the influence of active fires on the Biomass Burning Factor. While there
was evidence of active fires in North America and Eurasia occurring near a trajectory endpoint
within one hour, the uncertainty of a trajectory with a length of 336 hours is quite large (Stohl,
1998). Therefore, no meaningful conclusions can be drawn from this analysis other than the
transport time of emissions influencing the Biomass Burning Factor is greater than two weeks,
and that we are unable to capture these emissions with the current trajectory models with any
confidence."

4) Spatial origin of the Marine Cryosphere factor

Figure 6 (trajectory frequency) shows more frequent air masses from coastal regions but does
not show that these areas are responsible for enhanced marine cryosphere factor. It does not
support this sentence in the conclusion "Back trajectory analysis yielded MIZs around the
coasts of Greenland and the Arctic Ocean as source regions". I suggest a Potential Source
Contribution Function (PSCF) analysis to determine probable locations of emission sources.

Lines 443-454: see comment above on Figure 6 and the fact that it does not show that coastal
regions are responsible for enhanced marine cryosphere factor.

At the time of preparation of this manuscript, the authors did not possess the tools or
knowledge about how to perform a PSCF. The authors agree this would be the appropriate
method for determining source regions for the Marine Cryosphere Factor. Therefore, the
authors have become familiar with the programming language R and the R package Openair
(Carslaw and Ropkins, 2012). Using this package, the authors were able to produce a (PSCF)
for source region analysis of the Marine Cryosphere Factor. We have replaced the trajectory
frequency map for the summer season with a PSCF map using data from the entire campaign.
A PSCF for the summer period was also produced and compared to the entire campaign which
produced similar results. Inclusion of the entire campaign data provides a more robust
statistical calculation of the PSCF; therefore, we have chosen to perform the PSCF for the
entire campaign.

We have replaced the trajectory frequency map in Fig. 7 (previously Fig. 6, we have added a
figure showing the diurnal profile of the four factors in as the new Fig. 6 thus making this Fig.
7) with the PCSF as seen below and updated the figure caption accordingly.

[Figure]

Marine Cryosphere Factor

**Fig. 7.** PSCF for the Marine Cryosphere Factor and air mass back trajectories arriving at 100 m altitude, extending backward 120 hours in time. This plot and analysis method were produced in R and R Studio programs (R Foundation for Statistical Computing, Vienna, Austria, and R Studio Inc, MA, USA) and the OpenAir suite of analysis tools (Carslaw and Ropkins, 2012).

We have also updated Sect. 2.5 Back Trajectory Analysis to describe the PSCF:

Line 223–250: "To investigate source regions, the R package Openair (Carslaw and Ropkins, 2012) was utilized to produce a potential source contribution function (PSCF). Trajectories in Openair were calculated using the HYSPLIT model (Draxler and Hess, 1998; Rolph et al., 2017) at 100 m altitude and 120 hours backward in time using Global NOAA-NCEP/NCAR reanalysis data archives on a 2.5° resolution. A PSCF, shown in Eq. (3), calculates the probability that an emission source is located in a grid cell of latitude $i$ and longitude $j$, on the basis that emitted material in the gird cell $ij$ can be transported along the trajectory and reach the receptor site.

$$PSCF = \frac{m_{ij}}{n_{ij}} \tag{3}$$

Where $n_{ij}$ is the number of times a trajectory has passed through grid cell $ij$ and $m_{ij}$ is the number of times that a concentration was above a certain threshold value, in this case, the 90[th] percentile. To account for uncertainty in cells with a small number of trajectories passing through, a weighting function was applied (Carslaw and Ropkins, 2012).

We have amended the text in the Marine Cryosphere Factor section to reflect his new analysis method:

Line 527–541: "The spatial origin of the Marine Cryosphere Factor was investigated via a PSCF, calculated with the R package Openair (Carslaw and Ropkins, 2012). Figure 7 displays the PSCF for air masses arriving every hour during the measurement campaign, which provides increased statistical robustness to the results over calculating a PSCF just for the summer period. From Fig. 7, two areas with a high probability of being a source region for the Marine Cryosphere Factor can be discerned, the coast around Southeastern and Northeastern

Greenland. This analysis is supported by the CPF for the Marine Cryosphere Factor (Fig. S8b), which shows the dominant wind direction for this factor to be the south and south-south-east. Lee et al. (2020) used monthly chlorophyll-*a* derived from the MODIS satellite to demonstrate the coasts around Northeastern Greenland to contain high chlorophyll-*a* concentrations during June, which has been supported by previous studies (Degerlund and Eilertsen, 2010; Galí and Simó, 2010). Lee et al. (2020) also used a PSCF to determine this area to be the source regions for total particle number concentrations in the nucleation size range (3–25 nm). This area has been demonstrated to be a source region for MSA during the summer months (Heintzenberg et al., 2017). Thus, we propose the biologically active coasts around Eastern Greenland to be the source region for the Marine Cryosphere Factor."

Line-by-line comments:

5) Line 10: "we report a long-term dataset". The authors report measurement from April to October 2018, i.e., less than a year. This is not what I would call a "long-term dataset". Please edit this sentence.

We have replaced "long-term" with "multi-season" throughout the text to better reflect the duration of the dataset.

6) Line 33: Define VOCs and NOx.

 Line 35 and 36: "$NO_x$" has been defined as "nitrogen oxides" and "VOCs" have been defined as "volatile organic compounds".

7) Line 49: Define DMS.

Line 52: "DMS" has been defined as "dimethyl sulfide".

8) Line 108-109: "with low time resolution". Be more specific here. Gautrois et al. (2003) collected about one sample every 9 days. Additionally, the authors did not use a GC- MS, but a combination of GC-FID and GC-ECD.

The text has been amended to read:

Line 109–112: "Gautrois et al. (2003) reported long-term VOC concentrations for Alert, NU, where a seven-year time-series of VOCs mixing ratios has been generated, although with a 9 day time resolution, using off-line techniques (GC coupled to flame ionization and electron capture detectors)."

9) Line 110: You don't really explain why we need high time-resolved measurements of VOCs. Do you expect a high temporal variability? How about the global atmospheric watch reactive gases measurement network (Schultz et al., 2015) – Aren't these measurements enough?

Line 111: Same comment as above. In the previous sentence, you highlight the need for long-term measurements of VOCs in the Arctic. While a substantial contribution to the literature, you "only" report several months of data. You could perhaps emphasize more the high temporal frequency of your measurements.

The reviewer is correct in pointing out the lack of explanation for the need of high-time resolved measurement. We do expect a high temporal variability, especially in the summer when meteorological conditions can change rapidly. Schultz et al. (2015) is an important piece of literature, which highlights the need for high-time resolved measurements, therefore we have highlighted the need for high-time resolved measurements with the addition of the following paragraph:

Line 112–120: "High time resolution measurements are of vital importance for the study of Arctic atmospheric chemistry. For instance, diurnal studies can only be accomplished with a fast response instrument, as grab samples and time-integrated samples (i.e., adsorbent tubes) will not capture the variability on short enough time scales (de Gouw and Warneke, 2007). Understanding the effects of meteorological parameters on VOCs levels requires an instrument response which is shorter than the transient event being observed. Also, flux measurements can only be achieved through fast responding instrumentation (Müller et al., 2010). The study of short-lived compounds, such as reactive halogen species, and their interactions with VOCs is only possible on short timescales. Finally, global networks have highlighted the need for a quick turnaround in the delivery of atmospheric species for the validation of global atmospheric composition forecasting systems (Schultz et al., 2015)."

The reference de Gouw and Warneke, 2007, Müller et al., 2010, and Schultz et al., 2015 are new and have been added to the reference list.

10) Line 123: Did you filter data for local contamination? If so, how?

   Line 157: How exactly did you remove the influence of local pollution. What criteria did you use for wind speed and direction?

We have added Section 2 "Quality Control Procedure" in the Supplement, which describes how local pollution was identified and removed, as seen below.

SI Line 35– 52: **"Quality Control Procedure**

Data were quality controlled by analysis of PNSD, ozone, wind direction and speed, and internal activity logs. Local pollution at Villum can arise from activity around the measurement site (e.g., passenger vehicles, all-terrain vehicles, snowmobiles, and heavy machinery) as well as from activities from Station Nord (e.g., waste incineration, vehicular activity, and aircraft landing, idling, and take off). Internal activity logs of visits to the measurement building were used to highlight periods when human activity could affect the measurements, periods where VOC levels were elevated over background levels for the duration of the visit to the station were removed. Measurements of PNSD and ozone were analyzed, in tandem, for sharp and sudden increases in the ultrafine mode (< 100 nm) aerosol particles and concurrent sharp and sudden decreases in ozone, increases in ultrafine mode particles are indications of vehicular emissions while decreases in ozone results from its titration with nitrogen oxides. These periods were further inspected for wind direction and speed, with winds coming from due north at low speeds indicative of local pollution from Station Nord. All periods where local pollution was suspected of influencing the measurements were visually inspected by a panel of three persons, a consensus was required before data were removed. Data were also quality controlled for abnormal levels of instrumental parameters (i.e., E/N ratio, drift tube temperature, pressure, and voltage), periods with large deviations from nominal values were removed. Certain compounds (DMS, formic acid, and acetic acid) exhibited a slow return to nominal values after a blank than before, this issue was especially evident in the summer, these periods were removed. All quality control was performed on VOCs at a 5 s time resolution, data was removed before averaging to 30-minute means."

11) Line 125, Table S1: I have a hard time understanding how the seasons were defined. Skov et al. (2020) recently used a different (and more straightforward) definition: winter from December to February, spring from March to May, summer from June to August, and fall from September to November.

The authors admit this is an unusual set of dates for dividing seasons. This is because the data is split into three periods by interruptions (mainly due to power failure) as seen in Figure 1. Therefore, the authors divided the seasons according to these groups to include uninterrupted collected data. Additionally, "Fall" in the table has been changed to "Autumn", the manuscript has also been checked throughout for consistency regarding this naming. The slashes between measurement and unit has been removed and replaced with a comma. The dates have been made consistent with the dates listed in the manuscript. An updated Table 1 along with its caption is presented below:

**Table S1.** Statistics for meteorological parameters (mean ± s.d.) for all seasons, spring (April 4 – June 8), summer (June 9 – August 6), and autumn (August 7 – October 25). During the campaign, there were several large gaps in the data, most noticeably one in July and one in August, as seen in Fig. 1. The seasons are therefore divided based on the continuous collection of data uninterrupted by large missing gaps. The seasons roughly correspond to the conventional definition of seasons.

| | All Seasons | Spring | Summer | Autumn |
|---|---|---|---|---|
| Wind Direction, ° | 207.5 ± 89.0 | 202.4 ± 91.8 | 189.3 ± 2.6 | 223.8 ± 81.2 |
| Wind Speed, m s$^{-1}$ | 3.3 ± 2.6 | 3.1 ± 2.4 | 3.5 ± 2.4 | 3.4 ± 2.7 |
| Temperature, °C | -6.5 ± 9.6 | -13.8 ± 9.0 | 2.2 ± 4.1 | -7.0 ± 7.9 |
| RH, % | 77.4 ± 12.6 | 74.6 ± 10.6 | 78.0 ± 15.6 | 79.1 ± 11.4 |
| Radiation, W m$^{-2}$ | 174.9 ± 163.9 | 222.3 ± 146.3 | 295.9 ± 4.2 | 57.0 ± 97.4 |
| Pressure, hPa | 1010.6 ± 9.0 | 1014.8 ± 8.6 | 1007.5 ± 6.5 | 1009.6 ± 9.5 |
| Snow Depth, m | 0.9 ± 0.6 | 1.4 ± 0.1 | 1.1 ± 0.4 | 0.3 ± 0.4 |

12) Line 137-138: "measurements were interrupted for short periods ranging from days to weeks". Could you please add a Table summarizing, for each month, the number of hours of operation?

Table S2 has been added to the Supplement summarizing the number of hours the instrument was in operation for each compound for each month of the campaign. The following text has been added:

Line 151–152: "Table S2 summarizes the total number of operational hours for each compound for each month of the campaign." See Table 2 below.

**Table S2.** Total hours of operation of the PTR-ToF-MS for each month of the campaign and for each compound. Periods removed through the QC procedure are not included.

| | April | May | June | July | August | September | October |
|---|---|---|---|---|---|---|---|

| | | | | | | | |
|---|---|---|---|---|---|---|---|
| Formaldehyde | 374 | 601 | 288 | 661 | 417 | 443 | 403 |
| Acetonitrile | 229 | 601 | 288 | 661 | 417 | 443 | 403 |
| Formic Acid | 349 | 601 | 288 | 641 | 417 | 443 | 403 |
| Acetone | 376 | 601 | 288 | 661 | 417 | 443 | 403 |
| Acetic Acid | 375 | 577 | 288 | 661 | 417 | 411 | 359 |
| DMS | 300 | 577 | 169 | 391 | 357 | 443 | 377 |
| MEK | 376 | 601 | 288 | 661 | 417 | 443 | 403 |
| $C_3H_6O_2$ | 327 | 601 | 288 | 661 | 417 | 443 | 403 |
| Benzene | 376 | 601 | 288 | 661 | 417 | 443 | 403 |
| $C_5H_8O$ | 376 | 601 | 288 | 661 | 417 | 443 | 403 |

13) Line 144: "within the analytical uncertainties". Please refer to Table 1 here. Additionally, how often did you perform a calibration?

VOC mixing ratios were quantified using the kinetic rate reaction method (Supplement Sect. 1) and were validated against a certified reference standard at the beginning of the campaign. We have added a reference to (Holzinger et al., 2019) in the sentence, which refers to the quantification method and the reference standard. The phrase "using the reaction kinetics quantification method." has been added to the Supplement on Line 2. Table 1 has been referred to in the sentence. The main text was amended to:

Line 153: "Data generated by the PTR-ToF-MS instrument were processed with the PTR-MS Viewer software v. 3.2.12 (Ionicon Analytik). Mass calibrations and VOC mixing ratios were calculated by the PTR-MS Viewer, based on the reaction kinetics quantification method (Sect. S1). The instrument quantification was validated against an external gas-phase calibration standard (Apel-Riemer Environmental), a comparison between standard and instrument mixing ratios yielded percent errors that were within the analytical uncertainties (Table 1), therefore we are confident in the quantification method (Holzinger et al., 2019)."

14) Lines 179-181: Did you perform a sensitivity test? How does changing concentrations below LOD and missing concentrations influence the PMF results?

The authors performed an innumerable amount of PMF runs, varying treatment of data below the LOD, treatment of missing values (either removing the sample or replacing with median for the dataset), treatment of the uncertainty matrix, number of species included in the model (species were systematically removed and added), threshold values for species categorization, and number of factors. While each model run, produced unique results, the overall shape of the factor time series and species profile for each factor was consistent with the final reported model setup. The optimal model solution (as configured in the study) was deemed robust to these different variations of the dataset.

The text has been amended to include this description of model robustness.

Line 203–211: "Numerous sensitivity runs were performed to evaluate the validity of this data preparation protocol including varying the treatment of data below the LOD (replacing with half of the LOD or leaving as is), treatment of missing values (removing the sample or replacing missing species with the median), treatment of the uncertainty matrix, number of species included in the model (species were systematically removed or added to observe their influence on the model solution), threshold values for species categorization, and the number of factors.
Each variation of the input data, of course, produced a unique solution, however, the overall
shape of the time series and factor contributions profile was consistent with the solution present
in this study. The optimal model solution, for the configuration present here, was therefore
deemed robust to these variations of the input data and provided acceptable diagnostics."

15) Line 191: "automatic weather station placed close to the measurement site". How close?
Be more specific.

The text now states the distance of the automatic weather station from the measurement site.

Line 218–219: "Meteorological data including temperature, relative humidity, wind speed,
wind direction, pressure, radiation, and snow depth were generated by an automatic weather
station placed ~44 meters away from the measurement site."

16) Line 257: "Elevated DMS mixing ratios". What do you mean by "elevated"? Be more
specific.

The text has been amended to better describe the elevated mixing ratios of DMS during these
periods. The main text now reads:

Line 321: "Elevated DMS mixing ratios were observed for two short periods of a few days'
duration in May (1–5 May and 16–19 May), where DMS mixing ratios increased an order of
magnitude from ~0.02 to >0.2 ppbv (Fig. 3a and b)."

17) Line 273: "illustrated here by changes in wind speed". I would expect changes in wind
direction to be a more useful tracer of change in meteorological conditions.

The authors also expected wind direction to be more useful tracer of meteorological conditions,
however, for the two episodes of elevated DMS, changes in wind speed appear to be a better
indicator than wind direction. To reflect this, we have added wind direction to Fig. 3 in the
manuscript (see below). For the first episode, the wind direction is quite variable while
increased wind speeds are observed during depletions in acetone and elevations of DMS, and
it is unfortunate meteorological data are missing on 3rd and 4th of May. For the second episode,
the wind direction does change concurrently with an increase in wind speed, although
throughout the episode wind direction is also variable with contributions from the north and
the east. We have mentioned this in the text:

Line 338–339: "These changes in mixing ratios are accompanied by a change in meteorological
conditions, illustrated here by changes in wind speed and to a less extent wind direction (Fig.
3)."

[Figure]

**Fig. 3.** Left: The first period of elevated DMS mixing ratios (May 1–5). Right: The second period of elevated DMS mixing ratios (May 15– 19); (a) and (b) mixing ratios of acetone, DMS (left axis), and ozone (right axis); (c) and (d) wind speed (left axis) and radiation and wind direction (right axis). The shaded area represents episodes of elevated DMS mixing ratios.

18) Line 344: "a sink during the summer". A sink of what?

Line 414: "Increased areas of open water in the Arctic also act as a sink for acetonitrile during the summer (de Gouw et al., 2003)."

19) Lines 426-427: "Although, the variation of the Marine Cryosphere Factors seems not to be driven mainly by the dependence on horizontal wind speed (Fig. S2)". Figure S2 does not illustrate this. What is the correlation coefficient between the Marine Cryosphere Factor and wind speed?

The reference to Fig. S2 has been removed and replaced with the correlation coefficient between the Marine Cryosphere Factor and wind speed (R=-0.04) to better illustrate the lack of dependence between the two.

Line 509–510: "Although, the variation of the Marine Cryosphere Factor seems not to be driven mainly by the dependence on horizontal wind speed (R=-0.04)."

20) Lines 428-429: "given the distance of the measuring site from open water". What is the distance between the station and open water?

The fjord immediate to the station is located ~1.7 km away, during the summer this is mostly ice free, although is prone to freeze-ups when the temperature drops below zero for several hours. The station is located on a peninsula which is surrounded by sea ice throughout the year, taking this sea ice into account, open water is ~25 km away. The following text has been added:

Line 510–513: "Marine microorganisms produce DMS (Stefels et al., 2007; Levasseur, 2013), and given the distance of the measuring site from open water (taking sea ice into account the, station is approx. 25 km distance from open water), it is proposed that the majority of DMS produced is already oxidized to MSA and other products when reaching the station."

21) Section on Arctic Haze: please mention/discuss more clearly that you do not have data in wintertime, when Arctic Haze is expected to be at its maximum.

It has been made clear to the reader that our Arctic Haze Factor is only from spring and other studies present data from winter and spring. The following text has been added.

Line 617–620: "It is worth noting that the Arctic Haze Factor from this study is only for spring, while the other studies present data from the winter/spring, therefore any comparisons we make are from our spring Arctic Haze Factor to other Haze factors during winter and spring. While this is not a perfect comparison, it is one worth making, as Arctic Haze is the main source of anthropogenic pollution in the Arctic."

References

[revised manuscript text omitted]

---

## Author Comment (AC3) · 2 Dec 2020

Reply to *Interactive Comment* on "Atmospheric VOC measurements at a High Arctic site: characteristics and source apportionment" from anonymous Referee # 1

This manuscript reports on a long-term (spring through fall) Arctic VOC dataset observed at Villum Research Station at Station Nord in Greenland, and a PMF analysis performed on the dataset. The authors report the PTR-ToF-MS results for 10 detected ions, assigning 10 gas-phase molecular formulae and species/species groups to the observed ions in the PTR. A PMF analysis of the 10 species and species groups with a four-factor analysis is presented and discussed at length, including a Biomass Burning Factor, a combination Marine Cryosphere Factor, a Background Factor, and an Arctic Haze Factor. The authors give a very nice detailed analysis of the four factors, including the primary components, sources and influences and temporal variability.

We would like to thank referee # 1 for carefully reading the manuscript and for useful comments and feedback. We feel it improved the manuscript's readability and overall discussion of the results. As the first author is an early career scientist, they feel this exercise in the peer-review has tremendously helped them progress in critical thinking, manuscript writing, and the scientific method. We have addressed the referee's concerns and corrected errors in the manuscript below with referee's comments numbered and the author's responses in blue. New references are highlighted in yellow.

Several of the referee's concerns arose from the lack of explanation of the VOC specificity. We have group several of his comments into one and responded to them all with one reply.

1) One of my primary concerns with the paper, and with the majority of PTR- instrument papers, is that there is a lack of accounting or explanation of the VOC specificity (or lack thereof) of the PTR technique. The authors make no effort in this paper to discuss the interfering or additional species that may comprise each observed chemical formula that make up several of their measurements – e.g., propanal's contribution to the signal attributed to acetone, isobutanal's and butanal's contribution to the signal attributed to MEK – even to justify the omission of these species from the discussion with adequate explanation and literature references. As well, the authors' treatment of methyl acetate and propionic acid is to suggest that the contributions from each species (or other species that might contribute to the $C_3H_6O_2H^+$ signal) are un- known in Section 2.2, but then they attribute the signal to methyl acetate in the Biomass Burning Factor, and propionic acid in the Marine Cryosphere Factor, with no justification as to the reasons for the identification. The authors need to add commentary for the species identification, and justify the assumed VOCs under different conditions, or simply refer to the observations as a generic $C_3H_6O_2$ VOC group. Also, as detailed below, references to VOCs that comprise the $C_5H_8O$ observation should be clear that the measurement is not of an ion ($C_5H_8OH^+$ or $C_5H_8O^+$), but of the $C_5H_8O$ VOC group.

2) Lines 145-152 – the authors describe the method by which "compound names" are assigned to the nine protonated masses, including Pagonis et al. and references, which is reasonable, and a priori knowledge, which is not something that can be reference checked. I would argue that there is insufficient justification given to identifying the masses which ignore contributions from additional compounds that may be included in the concentrations measured. The authors write "Another compound ($C_4H_8OH^+$) was doubly assigned to propionic acid and methyl acetate.", but they likely meant to write C3H6O2H+, which has
m/z 75.058. However, they should explain here why they don't include ethyl formate or
hydroxyacetone as possible compounds at this mass.

3)  Line 337 – "it is a source of methyl acetate as well. . ." – the authors recognize that methyl
acetate could be contributing to the C3H6O2H+ signal, but by labeling it "propionic acid"
in Table 2 and Figures 1, 3, 5, 7, etc., the identity of the compound is muddied. If the
authors truly believe that the species is primarily propionic acid, then the presence of
methyl acetate would be unimportant. If they believe that it is indeed a mixture of the two
(or more) species, then this should be made clear whenever it is being referred to.

We recognize that the points made by the referee are correct and have thus modified the
manuscript, accordingly, adding a more detailed discussion of the possible and most likely
assignments of the detected masses to chemical species:

Line 160: "The PTR-MS technique allows to observe species with a proton-affinity higher than
water, this encompasses most VOCs found in the atmosphere with the important exception of
alkanes. It does not allow for the distinction between isomers to be made. Compound names
were assigned based on comparison with the libraries from the PTR-MS Viewer and Pagonis
et al. (2019), and references therein. Inspection of the mass spectrum yielded ten protonated
masses from which an empirical formula was calculated, and compound names were assigned
for nine masses, as discussed in Sect. 3.1."

The following paragraph has been added at the beginning of Sect. 3.1:

Line 252: "The ten selected masses monitored by the PTR-TOF-MS and their assignments to
species names are presented in Table 1. Assignments are made by choosing the most plausible
contributions to an observed mass but each measured ion may have contributions from several
different isomeric molecules. The assignment of masses in the table to protonated molecules
of formaldehyde, acetonitrile, formic acid, acetic acid, and benzene appears to be
unproblematic as no meaningful alternatives are found. For the remaining molecules,
alternative assignments are possible. The mass assigned to acetone could be propanal as well,
but propanal has a shorter atmospheric residence time and acetone is known to be one of the
dominating VOCs observed in the atmosphere (Jacob et al., 2002), further, it has been found
to have sources in the Arctic (Guimbaud et al., 2002). The mass assigned to DMS could be
ethanethiol as well, but the large marine source of DMS makes it the most plausible assignment.
Methyl ethyl ketone is isomeric with butenal, but being the second most abundant ketone in
the atmosphere with, among others, apparently an oceanic source (Brewer et al., 2020) it
appears to be the best assignment. $C_3H_6O_2$ may stem from propionic acid but also
hydroxyacetone, methyl acetate, and ethyl formate. While it seems unlikely that ethyl formate
could give a major contribution to this signal, the other three species are all plausible
candidates: Low molecular weight organic acids are commonly found in the atmosphere (Lee
et al., 2009), methyl acetate has been found in emissions from biomass burning (Andreae,
2019) and hydroxyacetone is known to be formed by the atmospheric degradation of isoprene
(Karl et al., 2009). For what concerns the $C_5H_8OH^+$ ion we prefer not to make an assignment,
possible isomers include, among others, pentenals and pentenones."

The references Jacob et al., 2002, Brewer et al., 2020,  Lee et al., 2009 and Karl et al., 2009 are new and have been added to the list of references.

Line 271 and lines 282–284: Sentences have been deleted.

Line 286: 'propionic acid' has been replaced by "$C_3H_6O_2$".

Lines 297–298: 'an oxidation product of n-butane' has been deleted.

Line 404–405: "one of the $C_3H_6O_2$ isomers" has been added to the sentence.

Line 462: 'propionic acid' has been replaced by "$C_3H_6O_2$".

Line 469–471: "The $C_3H_6O_2$ is in this case assigned to propionic acid as the alternative isomers seem less probable, considering their typical origins (biomass burning for methyl acetate and isoprene oxidation for hydroxyacetone)."

Line 564: 'propionic acid' has been replaced by "$C_3H_6O_2$".

The following sentence has been added:

Line 564–565: "$C_3H_6O_2$ may in this case result from all three of the isomers: propionic acid, methyl acetate, and hydroxyacetone."

Throughout the manuscript, and specifically in Figure 1, 2, 4, 5, 8, and 10 as well as Table 2, S2, S3, S4 and S5, 'propionic acid' has been replaced by "$C_3H_6O_2$".

4) My other primary concern is that the authors indicate that the data are available by contacting one of two author email addresses. I would strongly recommend that the paper not be published until the data are available in a publicly-available DOI.

The data for this manuscript including VOC mixing ratios and their associated uncertainty can be found in a publicly-available DOI. The Data Availability section has been amended to now read:

Line 673–675: All data used in this publication are available at https://doi.org/10.5281/zenodo.4299817 or by request to the corresponding authors Jakob Boyd Pernov (jbp@envs.au.dk) and Rossana Bossi (rbo@envs.au.dk).

Lines 155-157 – The authors should be specific about how the data were quality controlled using these parameters (PSND, WD, WS, etc.), and the resulting amount (total percentage, number of time periods, etc.) of data that had to be eliminated from the useful data set.

We have added Section 2 "Quality Control Procedure" in the Supplement which describes how local pollution was identified and removed (see text below). We have also a column in Table 1 which lists the total percentage of data removed due to QC (see an updated Table 1 below).

SI Line 35–52: **"Quality Control Procedure**

Data were quality controlled by analysis of PNSD, ozone, wind direction and speed, and internal activity logs. Local pollution at Villum can arise from activity around the measurement site (e.g., passenger vehicles, all-terrain vehicles, snowmobiles, and heavy machinery) as well as from activities from Station Nord (e.g., waste incineration, vehicular activity, and aircraft landing, idling, and take off). Internal activity logs of visits to the measurement building were used to highlight periods when human activity could affect the measurements, periods where
VOC levels were elevated over background levels for the duration of the visit to the station
were removed. Measurements of PNSD and ozone were analyzed, in tandem, for sharp and
sudden increases in the ultrafine mode (< 100 nm) aerosol particles and concurrent sharp and
sudden decreases in ozone, increases in ultrafine mode particles are indications of vehicular
emissions while decreases in ozone results from its titration with nitrogen oxides. These periods
were further inspected for wind direction and speed, with winds coming from due north at low
speeds indicative of local pollution from Station Nord. All periods where local pollution was
suspected of influencing the measurements were visually inspected by a panel of three persons,
a consensus was required before data were removed. Data were also quality controlled for
abnormal levels of instrumental parameters (i.e., E/N ratio, drift tube temperature, pressure,
and voltage), periods with large deviations from nominal values were removed. Certain
compounds (DMS, formic acid, and acetic acid) exhibited a slow return to nominal values after
a blank than before, this issue was especially evident in the summer, these periods were
removed.  All quality control was performed on VOCs at a 5 s time resolution, data was
removed before averaging to 30-minute means."

**Table 1.** Overview of measured protonated masses included in PMF analysis.  Mean refers to the arithmetic
average of the mixing ratio for each compound. Mean, Mean LOD, and % < LOD were calculated after quality
control of data influenced by local pollution. % QC represents the percentage of data removed due to the Quality
Control Procedure (Sect. S2).

| Measured mass ($m/z$) | Empirical Formula | Assigned Compound | Mean (ppbv) | Mean LOD (ppbv) | % < LOD | Mean Relative Uncertainty (%) | % QC |
|---|---|---|---|---|---|---|---|
| 30.997 | $CH_2OH^+$ | Formaldehyde | 0.220 | 0.176 | 0.6 | 41 | 5 |
| 42.019 | $C_2H_3NH^+$ | Acetonitrile | 0.067 | 0.045 | 0 | 46 | 5 |
| 47.011 | $CH_2O_2H^+$ | Formic Acid | 0.454 | 0.250 | 17 | 37 | 7 |
| 59.062 | $C_3H_6OH^+$ | Acetone | 0.608 | 0.037 | 0 | 32 | 0 |
| 61.047 | $C_2H_4O_2H^+$ | Acetic Acid | 0.201 | 0.096 | 5 | 39 | 8 |
| 63.034 | $C_2H_6SH^+$ | Dimethyl Sulfide | 0.046 | 0.043 | 4 | 57 | 25 |
| 73.068 | $C_4H_8OH^+$ | Methyl Ethyl Ketone | 0.031 | 0.023 | 0.1 | 56 | 0 |
| 75.058 | $C_3H_6O_2H^+$ | Propionic Acid / Hydroxyacetone/ Methyl Acetate | 0.025 | 0.031 | 0.1 | 61 | 2 |
| 79.057 | $C_6H_6H^+$ | Benzene | 0.027 | 0.031 | 0.5 | 64 | 0 |
| 85.066 | $C_5H_8OH^+$ | N/A | 0.027 | 0.030 | 0.03 | 61 | 0 |

5) Lines 215,  212,  467,  506,  etc.  – Technically  the authors did not observe ambient
C5H8O+ ions (or C5H8OH+ ions), but rather a compilation of [some] gas-phase C5H8O
species, which were protonated in order to be observed by the PTR system, similar to how
they did not observe atmospheric ambient C6H6H+ ions, but rather gas- phase C6H6 (i.e.,
benzene). Thus, discussion of the species or group of species with the chemical formula

C5H8O should simply be "C5H8O species" or "C5H8O", as in Figure 1, and should not
imply the measurement of an atmospheric ion.

The group of species identified at m/z 85.066 is now referred to as simply "$C_5H_8O$" throughout
the text.

6) Lines 245-252 – The comparisons presented against literature data from similar Arctic
stations make sense, for the most part, but the comparison of wintertime benzene mixing
ratios from Gautrois et al. (2003) to this study are not merited, as no winter time data is
being presented here. As well, while I agree that it has been shown that benzene and
acetonitrile are influenced by lower latitudes, the claim that acetonitrile is influenced by
anthropogenic emissions is not backed up. Remote levels of acetonitrile are likely impacted
by the significance of mid-latitude fire seasons, and are not expected to compare well from
year to year.

The referee is correct no wintertime data was collected. We compared our spring period
measurements to Gautrois et al. (2003) wintertime data. The authors agree this comparison
could create some confusion as it was not indicated in the text that we compared springtime to
wintertime data. The text had been amended to reflect only comparisons during summer.

Line 305–308: "Benzene has shown a seasonal pattern at Alert, NU with a higher mixing ratio
in winter due to no or limited photochemistry and long-range transport from lower latitudes
(Gautrois et al., 2003). They reported mean winter and summer mixing ratios of 0.200 and
0.034 ppbv, respectively; when compared to the present study there is good agreement during
the summer."

The authors interpreted the similar pattern as benzene during spring to be indicative of
anthropogenic influence, although the referee is correct, year to year variability from fires could
hinder the proper interpretation of this pattern. The text has been amended as follows:

Line 309–310: "Acetonitrile followed a similar pattern to benzene during the spring with
decreasing values, as well as exhibiting minima in the summer and maxima during the autumn
(Fig. 1b)."

We have added the following sentence:

Line 314: The main source of acetonitrile in the atmosphere has been found to be biomass
burning (Singh et al., 2003).

The reference Singh et al., 2003 is new and has thus been added to the list of references.

The discussion of acetonitrile during spring in Sect. 3.3.1 has also been changed:

Lines 407-412: "The decrease in the spring is reflective of decreasing concentrations of
benzene and acetonitrile; in the case of benzene this can be ascribed to anthropogenic emissions
during this period as the polar dome is expanded during winter and spring allowing for
emissions to be entrained from the mid-latitudes. In the case of acetonitrile, the reason is more
uncertain, there are anthropogenic sources of acetonitrile, particularly wood burning for
residential heating and solvent use (Languille et al., 2020), but they appear to be of very minor
importance compared to forest fires (de Gouw et al., 2003)."

Languille et al., 2020 is a new reference that has been added to the list of references.

We have also added the following text in the Arctic Haze section when we compare our Arctic Haze Factor to other Haze factors from previous literature.

Line 617–620: "It is worth noting that the Arctic Haze Factor from this study is only for spring, while the other studies present data from the winter/spring, therefore any comparisons we make are from our spring Arctic Haze Factor to other Haze factors during winter and spring. While this is not a perfect comparison, it is one worth making, as Arctic Haze is the main source of anthropogenic pollution in the Arctic."

7) Line 308 – the authors state that species with S/N < 0.2 were excluded from the analysis, but all 10 species (or species groups) discussed in the paper are included in Table 2. Are there any other species that were measured but not included here?

The species listed in Table 1 and 2 were the compounds identified that could be reasonably identified with an empirical formula with a proton affinity greater than water, without interference from neighboring ions, and exhibited a meaningful temporal profile.

The PTR measures ions with a m/z ratio up to 430 Da, so there are hundreds of ions measured by the instrument, but the ions reported here are the only those the authors could be confident were real signals from ambient VOCs. To answer the referee's question, no there was not.

8) Line 445 – The back trajectories frequency map for the Marine Cryosphere Factor is interesting, but it would be more informative to highlight some of the brief periods where this factor is particularly elevated, rather than averaging over a three-month summer period. Given, as well, that all the species identified to contribute to the Marine Cryosphere Factor have atmospheric lifetimes < 5 days, it would be prudent to limit these back trajectories to 120 hours or less.

The second referee has asked for a potential source contribution function (PSCF) for source region analysis of the Marine Cryosphere Factor. The authors agree this would be the appropriate method for determining source regions for the Marine Cryosphere Factor. Therefore, the authors have become familiar with the programming language R and the R package Openair (Carslaw and Ropkins, 2012). Using this package, the authors were able to produce a (PSCF) for source region analysis of the Marine Cryosphere Factor. We have replaced the trajectory frequency map for the summer season with a PSCF map for the entire campaign. A PSCF for the summer period was also produced and compared to the entire campaign which produced similar results. Inclusion of the entire campaign data provides a more robust statistical calculation of the PSCF; therefore, we have chosen to perform the PSCF for the entire campaign.

We have replaced the trajectory frequency map in Fig. 7 (previously Fig. 6, we have added a figure showing the diurnal profile of the four factors in as the new Fig. 6 thus making this Fig. 7) with the PCSF as seen below, and updated the figure caption accordingly.

[Figure]

**Fig. 7.** PSCF for the Marine Cryosphere Factor and air mass back trajectories arriving at 100 m altitude, extending backward 120 hours in time. This plot and analysis method were produced in R and R Studio programs (R Foundation for Statistical Computing, Vienna, Austria, and R Studio Inc, MA, USA) and the OpenAir suite of analysis tools (Carslaw and Ropkins, 2012).

We have also updated Sect. 2.5 Back Trajectory Analysis to describe the PSCF:

Line 232–250: "To investigate source regions, the R package Openair (Carslaw and Ropkins, 2012) was utilized to produce a potential source contribution function (PSCF). Trajectories in Openair were calculated using the HYSPLIT model (Draxler and Hess, 1998; Rolph et al., 2017) at 100 m altitude and 120 hours backwards in time using Global NOAA-NCEP/NCAR reanalysis data archives on a 2.5° resolution. A PSCF, shown in Eq. (3), calculates the probability that an emission source is located in a grid cell of latitude $i$ and longitude $j$, on the basis that emitted material in the gird cell $ij$ can be transported along the trajectory and reach the receptor site.

$$PSCF = \frac{m_{ij}}{n_{ij}} \qquad (3)$$

Where $n_{ij}$ is the number of times a trajectory has passed through grid cell $ij$ and $m_{ij}$ is the number of times that a concentration was above a certain threshold value, in this case the 90[th] percentile. To account for uncertainty in cells with a small number of trajectories passing through, a weighting function was applied (Carslaw and Ropkins, 2012)."

We have also added the following text in the Marine Cryosphere Factor section discussing the results.

Line 527–541: "The spatial origin of the Marine Cryosphere Factor was investigated via a PSCF, calculated with the R package Openair (Carslaw and Ropkins, 2012). Figure 7 displays the PSCF for air masses arriving every hour during the measurement campaign, which provides increased statistical robustness to the results over calculating a PSCF just for the summer period. From Fig. 7, two areas with a high probability of being a source region for the Marine Cryosphere Factor can be discerned, the coast around Southeastern and Northeastern

Greenland. This analysis is supported by the CPF for the Marine Cryosphere Factor (Fig. S8b), which shows the dominant wind direction for this factor to be the south and south-south-east. Lee et al. (2020) used monthly chlorophyll-*a* derived from the MODIS satellite to demonstrate the coasts around Northeastern Greenland to contain high chlorophyll-*a* concentrations during June, which has been supported by previous studies (Degerlund and Eilertsen, 2010; Galí and Simó, 2010). Lee et al. (2020) also used a PSCF to determine this area to be the source regions for total particle number concentrations in the nucleation size range (3–25 nm). This area has been demonstrated to be a source region for MSA during the summer months (Heintzenberg et al., 2017). Thus, we propose the biologically active coasts around Eastern Greenland to be the source region for the Marine Cryosphere Factor."

The references Carslaw and Ropkins, 2012, Degerlund and Eilertsen, 2010, Galí and Simó, 2010, Lee et al. (2020), and Heintzenberg et al., 2017 are new and thus have been added to the reference list.

9) Figures – all figures in the primary manuscript and supplement should be saved at a higher resolution. There is significant pixilation when zooming in on the plots. Some of the finer details are lost as a result, and some of the axis labels are rendered illegible.

All figures included in the manuscript have been saved at a resolution of 600 DPI. This is an excellent suggestion and in the future the authors will be more attentive to this matter.

10) Table 1 – The table title doesn't need to be so long. "Overview of measured protonated masses included in PMF analysis" would be sufficient. The rest is redundant with the table header, although Mean Mixing Ratio should be spelled out in the header or defined in a footnote. As well, it would be good to specify if the "Percentage below LOD" is the percentage of all data collected, or the percentage of only the data that was not removed due to the influence of local pollution. The same comment goes for the means reported.

The table title has been shortened with redundant information removed and the following text added:

Table 1: "Overview of measured protonated masses included in PMF analysis. Mean refers to the arithmetic average of the mixing ratio for each compound. Mean, Mean LOD, and % < LOD were calculated after quality control of data influenced by local pollution. % QC represents the percentage of data removed due to the Quality Control Procedure (Sect. S2)."

11) Line 27 – "rate" would be preferable to "speed".

Line 29: "Speed" has been replaced with "rate".

12) Line 33 – NOx should be defined.

Line 35 and 36: "$NO_x$" has been defined as "nitrogen oxides" and "VOCs" have been defined as "volatile organic compounds".

Line 52: "DMS" has been defined as "dimethyl sulfide".

13) Line 46 – there is a rogue hyphen/em dash that isn't needed.

Line 48: The rouge em dash has been removed.

14) Line 46 and others – Dall'Osto is missing an apostrophe both here and in the refer- ence list, where the reference is also missing several other diacritical marks, and the majority of C.D. O'Dowd's last name. The references should then be rearranged for this reference to come before the more recent Dall'Osto et al. references. Be wary of automatic reference management software – references should still be verified that they were transposed and recorded properly.

This is an excellent catch by the referee. The Dall'Osto reference has been corrected and the entire reference list has been checked for accuracy and updated where appropriate. This is excellent advice from the referee. We believe the problem arose from importing references from PDFs using the "Import" function in Endnote. We have now imported references either form Web of Science or the respective journal website. The authors were unaware of such pitfalls when working with automatic referencing software and will be more vigilant in the future.

References – Please format all references properly: pay attention to things like consistent journal abbreviations, consistent DOI referencing, missing or n/a information (e.g., line 680), line wrapping (e.g., line 735), and capitalization of abbreviations and proper names (e.g., lines 632; 839; 842, etc.).

15) Line 68 – "loss" rather than "reactions" would generate better agreement with the singular "sink".

Line 70: "reactions" has been replaced with "loss".

16) Lines 90-93 – "Furthermore, Boudries et al. (2002) observed emission from the snow- pack to the atmosphere of acetone, acetaldehyde, and formaldehyde, which were explained by photochemical production in the snowpack and depositional fluxes of methanol was also observed, which they postulated as a source of formaldehyde."- Consider making this two sentences: "Furthermore, Boudries et al. (2002) observed emission from the snowpack to the atmosphere of acetone, acetaldehyde, and formaldehyde, which were explained by photochemical production in the snowpack. "Depositional fluxes of methanol were also observed, which they postulated as a source of formaldehyde" Or at the very least, add a semi-colon and change "was" to "were" on line 92.

Line 94: This sentence has been made into two sentence following the referee's suggestions and "was" is now "were".

17) Line 94 – there should be a comma after "VOCs".

Line 96: A comma has been added after VOCs.

18) Line 103 – It would be good to mention that Barrow, AK is now Utqiagvik, AK.

Throughout the text "Barrow" has been replaced with "Utqiagvik".

On line 105, it is mentioned that Utqiagvik is formerly known as Barrow.

19) Line 104 and others – While "Alert, CA" is technically acceptable, "Alert, Canada,", or "Alert, NU," would be less ambiguous. Also, be consistent throughout. Greenland should probably be spelled out as well.

Throughout the text "Alert, CA" has been replaced with "Alert, NU".

20) Line 122 – Use "s" instead of "sec" to adhere to SI units requirement.

Line 133: "sec" has been replaced with "s". The entire manuscript and SI has also been checked for proper use of SI units were appropriate.

21) Line 122 – Use "southwest" instead of "south-west".

Line 133: The hyphen has been removed.

22) Line 131 – "5 seconds scan rate" doesn't describe a rate, which should be something per unit of time.

Line 142: "5 seconds scan rate" has been replaced with "5 second single spectra integration time" as specified in the PTR software.

23) Line 154 – "mixing ratios below LOD were set to"

Line 173: "was" has been replaced with "were".

24) Line 155 – "the data were time-averaged to 30-minute means."

Line 173: "was" has been replaced with "were" and "mean" was made plural.

25) Line 194 – "arriving from"

Line 222: "form" has been replaced with "from".

26) Lines 199-200 – "Active fires during the period 15 August – 15 September 2018 were provided..." (you are defining the period here, not referring to it, so the commas are not needed.)

This sentence has been removed from this section and moved to the Biomass Burning section (Line 424), where the commas have been removed.

27) Line 259 – I recommend splitting this long sentence, ". . . frozen sea surface. Back trajectory. . ."

Line 324–325: The sentence has been split into two following the referee's suggestion.

28) Line 267 – "strong negative correlation" is a little too generous for R = -0.68.

Line 332–333: "strong" has been replaced with "moderate".

29) Lines 271-273 – It would be informative to include wind direction in addition to wind speed in Figure 2.

Wind direction has been added to Figure 3, which is the old Figure 2 after addition of a figure showing the diurnal profile of certain VOCs during the summer as suggested by the second referee. Wind direction has also been added to Fig. S2. The effect of wind direction has been included in the text:

Line 338: These changes in mixing ratios are accompanied by a change in meteorological conditions, illustrated here by changes in wind speed and to a less extent wind direction (Fig. 3).

369 30) Line 275 – "with elevated acetone levels during ozone. . ." or something similar.

370 The text has been amended following the referee's suggestion.

371 Line 339–341: Guimbaud et al. (2002) found a similar relationship between acetone and ozone
372 during different field campaigns at Alert, Canada with elevated acetone levels during ozone
373 depletion episodes accompanied by a concomitant decrease in the propane mixing ratios.

374 31) Line 279 – "gas-phase"

375 Line 342: A hyphen has been added to "gas-phase".

376 32) Line 303 – "species with S/N. . ."

377 Line 369: "Signal-to-noise" has been removed.

378 33) Line 304 – "The uncertainties of 'Weak' species were tripled. . ."

379 Line 370: "Uncertainty" has been made plural to "uncertainties" and "was" replaced with
380 "were".

381 34) Line 314 – "VOCs devoid of episodic influence. . .", and there is a period missing at the
382   end of the sentence.

383 Line 381: "void" has been replaced with "devoid" and a period has been added to the end of
384 this sentence.

385 35) Line 394 – The authors write "Estimated globally averaged atmospheric lifetimes against
386   wet deposition for formic and acetic acid in the boundary layer is between 1 and 2 days
387   respectively (Paulot et al., 2011)." – This is not clear. Are both of the estimated atmospheric
388   lifetimes between 1 and 2 days? If so, "respectively" isn't needed. Either way, though, it
389   should state "are between". . .

390 The text has been amended in the following manner:

391 Line 476–478: "Estimated globally averaged atmospheric lifetimes against deposition for both
392 formic and acetic acid in the boundary layer are between 1 and 2 days (Paulot et al., 2011)."

393 36) Line 396 – "14C" (with a superscripted 14) or "carbon-14" (without a superscript).

394 Line 479: A superscripted 14 has been added to the front of C, the text now reads "$^{14}$C".

395 37) Lines 405, 407, 410, 412, 443, 446, Figure 5, etc. etc. – sometimes "Factor" is capitalized
396   in reference to one of the four factors, and sometimes it isn't. This should be consistent
397   throughout.

398 The text has been amended throughout, when referring to a specific factor, "Factor" is now
399 capitalized.

400 38) Line 427 – "Factor", not "Factors". Also, there is an extra period in this sentence: ". . .
401   speed (Fig. S2.)."

402 Line 510: "Factors" is now singular "Factor" and the extra period has been removed and
403 reference to Fig. S2 has been removed and replaced with the correlation coefficient between
404 the Marine Cryosphere Factor and wind speed (as requested by the second referee).

39) Lines 430-431 – Despite what the papers might claim, MSA is not measured in particle
phase, but rather they measured the methanesulfonate ion, CH3SO2+. It would be better to
simply indicate that the presence of gas-phase MSA has been indicated by the observation
of methanesulfonate ion in particles.

The text has been amended in the following manner:

Line 513–515: "The presence of gas-phase MSA has been indicated by the observation of the
methanesulfonate ion, which has been previously measured in the particle phase at Villum in
February–May 2015 (Dall'Osto et al., 2018; Nielsen et al., 2019)."

40) Line 438 – ". . . Dibb and Arsenault (2002) measured levels. . ."

Line 522: The word "had" has been removed.

41) Line 440 – "matter, e.g.,"

Line 524: A comma has been added after "matter".

42) Line 444 – The sentence "These trajectories and trajectory frequency maps were cal-
culated as described in Sect. 2.4." isn't necessary.

Line 529–530: This sentence has been removed.

43) Lines 460-461 – recommend: "One of the source areas identified in Fig. 6 is southeast of
Villum, and a CPF analysis indicated high contributions (of what?) were observed when
the winds were from south of Villum (Fig. S8a)." – this sentence needs a little clean-up for
readability and clarity.

This sentence has been amended in the following manner:

Line 554–556: "One of the source areas identified in Fig. 7 is southeast of Villum, and a CPF
analysis indicated high contributions of the Marine Cryosphere Factor were observed when the
wind direction was south of Villum (Fig. S5b)."

44) Line 469 – Recommend changing "Most of its components, particularly acetone and
formaldehyde, are known. . ." to simply "Acetone and formaldehyde are known. . ."

Line 565: The sentence has been amended following the referee's suggestion.

45) Lines 483, 484, 487, 508, 545 – do you mean "labile [organic] carbon"?

Throughout the text "liable carbon" has been replaced with "labile organic carbon".

46) Line 531 – Circle should be capitalized.

Line 632: Circle is now capitalized.

47) Figure 4 – "red stars" – the resolution doesn't merit calling these stars. They're mostly just
dots.

This figure has been removed from the manuscript. The second referee asked for a more
statistical analysis of the back trajectories with the active fires. We collocated back trajectory
endpoints with active fires with 1° latitude/longitude and temporally within one hour. While
there was evidence of active fires in North America and Eurasia occurring when an endpoint was near, the uncertainty in individual trajectories at 336 hours is too great to draw meaningful
conclusions from this analysis.

We have included the figure in our response, this figure will not be included in the manuscript.
Individual trajectories are indicated in the dashed blue lines and active fires occurring within
1° lat/lon and within one hour of trajectory endpoints are indicated in red.

[Figure]

We have amended the text for the Biomass Burning section.

Line 413: To examine the geographical origin of this factor, air mass back trajectories from the
HYSPLIT model were calculated every hour during the peak of the Biomass Burning Factor
(15 August–15 September 2018) and extending 336 hours (two weeks) backward in time. The
trajectory length of two weeks was selected to account for the long lifetime of acetonitrile.
Active fires during the period 15 August–15 September 2018 was provided by NASA's Fire
Information for Resource Management System (FIRMS) (Schroeder et al., 2014). Active fires
occurring with one hour and one-degree latitude/longitude of a trajectory endpoint was used to
access the influence of active fires on the Biomass Burning Factor. While there was evidence
of active fires in North America and Eurasia occurring near a trajectory endpoint with one hour,
the uncertainty of a trajectory with a length of 336 hours is quite large (Stohl, 1998). Therefore,
no meaningful conclusions can be drawn from this analysis other than the transport time of
emissions influencing the Biomass Burning Factor is greater than two weeks, and that we are
unable to capture these emissions with the current trajectory models with any confidence.

Supplement

48) Line 26 – Either "(5 s)" or "(5 seconds)" would be acceptable SI units.

SI Line 27: "sec" has been changed to "s".

49) Table S1 - The way the authors divided up the seasons here seems oddly arbitrary. Why is
"summer" only two months long, while autumn is three months? And changing seasons on
the 7th of a month is oddly arbitrary. As well, it would be preferable to separate the
measurement and units in the first column with a comma rather than a slash. Also, use
either "autumn" or "fall" but not both in the table title and header. Lastly, the start and stop dates in the title are not consistent with the dates given on Line 128 of the main text. Please
make these consistent.

The authors admit this is an unusual set of dates for dividing seasons. This is because the data is split into three periods by interruptions (mainly due to power failure) as seen in Figure 1. Therefore, the authors divided the seasons according to these groups. "Fall" in the table has been changed to "Autumn", the manuscript has also been checked throughout for consistency regarding this naming. The slashes between measurement and unit has been removed and replaced with a comma. The dates have been made consistent with the dates listed in the manuscript. See the updated Table 1 below.

**Table S1.** Statistics for meteorological parameters (mean ± s.d.) for all seasons, spring (April 4 – June 8), summer (June 9 – August 6), and autumn (August 7 – October 25). During the campaign, there were several large gaps in the data, most noticeably one in July and one in August, as seen in Fig. 1. The seasons are therefore divided based on the continuous collection of data uninterrupted by large missing gaps. The seasons roughly correspond to the conventional definition of seasons.

|  | All Seasons | Spring | Summer | Autumn |
|---|---|---|---|---|
| Wind Direction, ° | 207.5 ± 89.0 | 202.4 ± 91.8 | 189.3 ± 2.6 | 223.8 ± 81.2 |
| Wind Speed, m s$^{-1}$ | 3.3 ± 2.6 | 3.1 ± 2.4 | 3.5 ± 2.4 | 3.4 ± 2.7 |
| Temperature, °C | -6.5 ± 9.6 | -13.8 ± 9.0 | 2.2 ± 4.1 | -7.0 ± 7.9 |
| RH, % | 77.4 ± 12.6 | 74.6 ± 10.6 | 78.0 ± 15.6 | 79.1 ± 11.4 |
| Radiation, W m$^{-2}$ | 174.9 ± 163.9 | 222.3 ± 146.3 | 295.9 ± 4.2 | 57.0 ± 97.4 |
| Pressure, hPa | 1010.6 ± 9.0 | 1014.8 ± 8.6 | 1007.5 ± 6.5 | 1009.6 ± 9.5 |
| Snow Depth, m | 0.9 ± 0.6 | 1.4 ± 0.1 | 1.1 ± 0.4 | 0.3 ± 0.4 |

50) Tables S2-S4 – It is unclear why June, July and September are included here, but not August and October. In the text, Villum Research Station is referred to as "Villum", not VRS. It should be the same here, or spelled out in full. The vertical alignment of these tables is off, with the numbers right justified, and the headers left-justified, making it difficult to know which values go with which headers. As well, some of the compounds listed in the left-hand column blend together. Either increase the spacing, or shorten the names (i.e., MEK, DMS, etc.) to limit the amount of word-wrapping. Formic Acid across the head is also rather unfortunately split. Finally, the "All correlations, apart from . . ." in the titles should just be included as a footnote.

The correlation analysis was performed for one month from each season which had a good data coverage for the parameters being compared. We have added Table S2, which details the number of measurement hours for each compound for each month that displays this. VRS has been changed to Villum in the table headers. The columns of Tables S3, S4, and S5 are all now left aligned. DMS and MEK are now used in the left column and top row, which eliminated all word-wrapping. The text "All correlations…" has been made a footnote. Please see the updated tables in the SI, considering the length of the tables they are not included here.

51) Figure S1 – the text suggests that there were times when the wind speeds were < 2 m/s, but this is not included in the figure. Please either include these, or justify their omission. Also, the resolution on the figure does not allow for the reader to discern anything > 14-18 m/s (blue). Either improve the resolution, or change the legend to eliminate the highest wind
speed categories.

The figure has been remade to include all wind speeds and the intervals of the color bar have
been changed to allow the relative wind speeds to be discerned. The figure has been expanded
for individual wind roses for each season. The figure was also saved at a higher resolution (300
vs 600 DPI). See the updated Fig. S1 below.

[Figure]

**Fig. S1.** Wind Rose for mean wind speed at 5 min time resolution for (a) all seasons, (b) spring,
(c) summer, and (d) autumn. The y-axis represents the percent frequency of wind direction in
percent and the colors indicate mean wind speed in m s$^{-1}$. The seasons follow the selection
outlined in Table 1.

52) Figure S2 – "Time series of meteorological parameters. . ."; consider adding wind di-
rection to this figure as well.

Wind direction has been added to this figure. The figure was also saved at a higher resolution
(300 vs 600 DPI). Please see the updated Fig. S2 in the Supplement.

53) Figure S4 (and S5) – there is a lot of information shown that is repetitive and unneeded to
the right of each satellite image, and as a result the majority of the important de- tails are
illegible. Remove the unnecessary parts, and make higher res and/or larger versions of the
plots, and label the leads and the station in the image(s). As well, the labels a-f should be
moved to the top left, or top right, or could be included inside the images in white for
clarity. Lastly, here and throughout the manuscript, re: the ACP style guide, dates should
be in the form dd month yyyy (or simply dd month).

The old Figures S4 and S5 have been removed from the manuscript. Both referees raised
concerns about the legibility of these two figures, therefore, we have removed them and
directed the reader to the website where they were obtained (Line 324). We feel they add
valuable information about the origin of the elevated DMS periods but displaying them in a
meaningful manner proved problematic.

Throughout the manuscript, texts and figures have been amended to display the correct date
format for ACP.

54) Figure S6 – caption "A new trajectory was [calculated/generated] every 24 hours." The
back trajectory trace colors in the plots should have a legend or be described.

The Fig. S6 caption now reads:

"**Fig. S6.** HYSPLIT back trajectory analysis for **(a)** May 2$^{st}$ – 6$^{th}$ **(b)** May 16$^{th}$–20$^{th}$ arriving at 100 m above ground level extending 72 hours backward in time. The colored trajectories represent a new trajectory started every 24 hours after the last day of each period until the first day, in descending order the trajectories are red (last day), blue (fourth day), green (third day), light blue (second day), and purple (first day)."

The figure was also remade at a higher resolution and with panel labels ((a) and (b)) located at the top left of each panel.

55) Figure S7 – The caption should include the fact that this is from the PMF analysis.

The caption for Fig. S7 now reads:

"The ratio of Q$_{true}$ to Q$_{theo}$ versus the number of factors for the PMF analysis."

The caption for Fig. S8 now reads:

"Conditional probability function roses for **(a)** Biomass Burning Factor, **(b)** Marine Cryosphere Factor, **(c)** Background Factor, and **(d)** Arctic Haze Factor."

56) Figure S8 – plots (a) and (c) have the same size CPF scale, but different numbers of ticks and significant figures. They should be the same.

Figure S8 has been updated to include all factors. The Biomass Burning, Background, and Arctic Haze Factors now all have the same scale, and all panels now have the same number of significant figures for the scale.

**References**

[revised manuscript text omitted]

---

## Author Response (AR2)

Reply to Reviewer Report #1 on "Atmospheric VOC measurements at a High Arctic site: characteristics and source apportionment"

We would like to thank the editor for their useful comments that will improve the manuscript, both in terms of clarity and scientific quality. We have addressed the editor's comments below with comments numbered and the author's responses in blue.

Non-public comments to the Author:
------ Reviewer Report #1 ------
I want to thank the authors for the comprehensive revisions of a very interesting study. Please consider the very minor comments below to further improve the manuscript in key places.

1. In the caption of Figure S1, the authors refer to Table 1 for the definition of the seasons. They should refer to Table S1 instead.

The text has been amended to refer to Table S1 in the caption of Figure S1. Thank you for this good catch.

2. According to the PSCF analysis, the probability of a source located on the coast is pretty low (~9 %). I'd suggest rephrasing as follows:
"From Fig. 6, two areas with a relatively higher probability of being a source region…"
Note that you can change the limits in the trajLevel function (let's say from 0 to 100 %) by including limits = c(0,1).

The authors agree this change will help in the interpretation of the PSCF results. The text has been amended according to the suggested changes.

Line 531–534: From Fig. 7, two areas with a relatively higher probability of being a source region for the Marine Cryosphere Factor can be discerned, the coast around Southeastern and Northeastern Greenland.

The authors are also grateful for the helpful tips regarding the trajLevel function. We have changed the scaling of Fig. 7 to a max of 0.12 so the potential source regions are not overly accentuated.

3. Thank you for including figures in the supplementary describing the diurnal cycle. I'm being picky but could you please include the 95% confidence intervals in the mean? This can easily be done using function timeVariation in R package openair.

The authors originally produced the figures with the 25th / 75th percentiles; however, their inclusion masked the shape of the diurnal profile therefore they were not included in the revised figures. The authors agree the 95 % confidence intervals are an excellent method to display a variance around the diurnal profiles while still display the shape of the profile. The figures have been updated to include these intervals and the captions have been updated to reflect this change.

[revised manuscript text omitted]

**2. Quality Control Procedure**

Data were quality controlled by analysis of PNSD, ozone, wind direction and speed, and internal activity logs. Local pollution at Villum can arise from activity around the measurement site (e.g., passenger vehicles, all-terrain vehicles, snowmobiles, and heavy machinery) as well as from activities from Station Nord (e.g., waste incineration, vehicular activity, and aircraft landing, idling, and take off). Internal activity logs of visits to the measurement building were used to highlight periods when human activity could affect the measurements, periods where VOC levels were elevated over background levels for the duration of the visit to the station were removed. Measurements of PNSD and ozone were analyzed, in tandem, for sharp and sudden increases in the ultrafine mode (< 100 nm) aerosol particles and concurrent sharp and sudden decreases in ozone, increases in ultrafine mode particles are indications of vehicular emissions while decreases in ozone results from its titration with nitrogen oxides. These periods were further inspected for wind direction and speed, with winds coming from due north at low speeds indicative of local pollution from Station Nord. All periods where local pollution was suspected of influencing the measurements were visually inspected by a panel of three persons, a consensus was required before data were removed. Data were also quality controlled for abnormal levels of instrumental parameters (i.e., E/N ratio, drift tube temperature, pressure, and voltage), periods with large deviations from nominal values were removed. Certain compounds (DMS, formic acid, and acetic acid) exhibited a slow return to nominal values after a blank than before, this issue was especially evident in the summer, these periods were removed.  All quality control was performed on VOCs at a 5 s time resolution, data was removed before averaging to 30-minute means.

**Table S1.** Statistics for meteorological parameters (mean ± s.d.) for all seasons, spring (April 4 – June 8), summer (June 9 – August 6), and autumn (August 7 – October 25). During the campaign, there were several large gaps in the data, most noticeably one in July and one in August, as seen in Fig. 1. The seasons are therefore divided based on the continuous collection of data uninterrupted by large missing gaps. The seasons roughly correspond to the conventional definition of seasons.

| | All Seasons | Spring | Summer | Autumn |
|---|---|---|---|---|
| Wind Direction, / ° | 207.5 ± 89.0 | 202.4 ± 91.8 | 189.3 ± 2.6 | 223.8 ± 81.2 |
| Wind Speed, / m s$^{-1}$ | 3.3 ± 2.6 | 3.1 ± 2.4 | 3.5 ± 2.4 | 3.4 ± 2.7 |
| Temperature, / °C | -6.5 ± 9.6 | -13.8 ± 9.0 | 2.2 ± 4.1 | -7.0 ± 7.9 |
| RH, / % | 77.4 ± 12.6 | 74.6 ± 10.6 | 78.0 ± 15.6 | 79.1 ± 11.4 |
| Radiation, / W m$^{-2}$ | 174.9 ± 163.9 | 222.3 ± 146.3 | 295.9 ± 4.2 | 57.0 ± 97.4 |
| Pressure, / hPa | 1010.6 ± 9.0 | 1014.8 ± 8.6 | 1007.5 ± 6.5 | 1009.6 ± 9.5 |
| Snow Depth, / m | 0.9 ± 0.6 | 1.4 ± 0.1 | 1.1 ± 0.4 | 0.3 ± 0.4 |

Formatted Table

**Table S2.** Total hours of operation of the PTR-ToF-MS for each month of the campaign and for each compound. Periods removed through the QC procedure are not included.

| | April | May | June | July | August | September | October |
|---|---|---|---|---|---|---|---|
| Formaldehyde | 374 | 601 | 288 | 661 | 417 | 443 | 403 |
| Acetonitrile | 229 | 601 | 288 | 661 | 417 | 443 | 403 |
| Formic Acid | 349 | 601 | 288 | 641 | 417 | 443 | 403 |
| Acetone | 376 | 601 | 288 | 661 | 417 | 443 | 403 |
| Acetic Acid | 375 | 577 | 288 | 661 | 417 | 411 | 359 |
| DMS | 300 | 577 | 169 | 391 | 357 | 443 | 377 |
| MEK | 376 | 601 | 288 | 661 | 417 | 443 | 403 |
| $C_3H_6O_2$ | 327 | 601 | 288 | 661 | 417 | 443 | 403 |
| Benzene | 376 | 601 | 288 | 661 | 417 | 443 | 403 |
| $C_5H_8O$ | 376 | 601 | 288 | 661 | 417 | 443 | 403 |

Formatted Table

**Table S32:** Pearson correlation coefficients[a] for chemical species, temperature and sun radiation measured during April at Villum.

| April 2018 | Formal-dehyde | Aceto-nitrile | Formic Acid | Acetone | Acetic Acid | DMS | MEK | Benzene | Propionic Acid | Tempe-rature | Radiation | Ozone |
|---|---|---|---|---|---|---|---|---|---|---|---|---|
| Formaldehyde | 1.00 | | | | | | | | | | | |
| Acetonitrile | 0.70 | 1.00 | | | | | | | | | | |
| Formic Acid | 0.76 | 0.45 | 1.00 | | | | | | | | | |
| Acetone | 0.40 | 0.30 | *-0.03* | 1.00 | | | | | | | | |
| Acetic Acid | -0.63 | -0.74 | -0.45 | -0.32 | 1.00 | | | | | | | |
| DMS | -0.47 | -0.67 | -0.16 | -0.55 | 0.84 | 1.00 | | | | | | |
| MEK | 0.52 | 0.20 | 0.76 | 0.03 | -0.27 | -0.07 | 1.00 | | | | | |
| Benzene | 0.27 | 0.04 | 0.70 | -0.43 | *-0.07* | 0.24 | 0.84 | 1.00 | | | | |
| Propionic Acid /  | -0.52 | -0.66 | -0.25 | -0.41 | 0.90 | 0.94 | -0.15 | 0.11 | 1.00 | | | |
| Temperature | -0.47 | -0.34 | -0.75 | 0.16 | 0.54 | 0.23 | -0.74 | -0.77 | 0.46 | 1.00 | | |
| Radiation | -0.26 | -0.26 | -0.38 | 0.28 | 0.20 | *0.06* | -0.25 | -0.34 | 0.21 | 0.34 | 1.00 | |
| Ozone | -0.52 | -0.48 | -0.21 | -0.83 | 0.56 | 0.64 | -0.26 | 0.15 | 0.59 | 0.17 | -0.12 | 1.00 |

70    ᵃ All correlations, apart from the numbers typed in italics, have linear regression p-values below 0.01.

**Table S4**: Pearson correlation coefficients[a] for chemical species, temperature and sun radiation measured during July at Villum.

| July 2018 | Formal-dehyde | Aceto-nitrile | Formic Acid | Acetone | Acetic Acid | DMS | MEK | Benzene | C₃H₆O₂ / Methyl Acetate | Tempe-rature | Radiation | Ozone |
|---|---|---|---|---|---|---|---|---|---|---|---|---|
| Formaldehyde | 1.00 | | | | | | | | | | | |
| Acetonitrile | 0.71 | 1.00 | | | | | | | | | | |
| Formic Acid | 0.88 | 0.57 | 1.00 | | | | | | | | | |
| Acetone | 0.86 | 0.89 | 0.82 | 1.00 | | | | | | | | |
| Acetic Acid | 0.85 | 0.58 | 0.95 | 0.85 | 1.00 | | | | | | | |
| DMS | 0.36 | *0.01* | 0.50 | 0.23 | 0.42 | 1.00 | | | | | | |
| MEK | 0.85 | 0.55 | 0.93 | 0.81 | 0.97 | 0.41 | 1.00 | | | | | |
| Benzene | 0.57 | 0.50 | 0.50 | 0.61 | 0.59 | 0.26 | 0.60 | 1.00 | | | | |
| C₃H₆O₂ / Methyl Acetate | 0.83 | 0.57 | 0.95 | 0.82 | 0.97 | 0.39 | 0.95 | 0.50 | 1.00 | | | |
| Temperature | 0.65 | 0.85 | 0.54 | 0.82 | 0.58 | *0.08* | 0.54 | 0.45 | 0.54 | 1.00 | | |
| Radiation | 0.49 | 0.23 | 0.59 | *0.40* | 0.51 | 0.26 | 0.53 | 0.15 | 0.56 | 0.31 | 1.00 | |
| Ozone | 0.54 | 0.82 | 0.39 | *0.69* | 0.39 | 0.18 | 0.33 | 0.43 | 0.33 | 0.76 | 0.07 | 1.00 |

[a] All correlations, apart from the numbers typed in italics, have linear regression p-values below 0.01.

**Table S54**: Pearson correlation coefficients a coefficients for chemical species, temperature and sun radiation measured during September at VillumRS. All correlations, apart from the numbers typed in italics, have linear regression p-values below 0.01.

| September 2018 | Formaldehyde | Acetonitrile | Formic Acid | Acetone | Acetic Acid | DMS | MEK | Benzene | $C_3H_6O_2$ | Temperature | Radiation | Ozone |
|---|---|---|---|---|---|---|---|---|---|---|---|---|
| Formaldehyde | 1.00 | - | - | - | - | - | - | - | - | - | - | - |
| Acetonitrile | 0.61 | 1.00 | - | - | - | - | - | - | - | - | - | - |
| Formic Acid | 0.76 | 0.45 | 1.00 | - | - | - | - | - | - | | - | |
| Acetone | 0.72 | 0.96 | 0.57 | 1.00 | - | - | - | - | - | - | - | |
| Acetic Acid | *0.06* | 0.29 | *0.07* | 0.28 | 1.00 | - | - | - | - | - | - | - |
| DMS | -0.29 | -0.76 | -0.18 | -0.68 | -0.10 | 1.00 | - | - | - | - | - | |
| MEK | 0.82 | 0.71 | 0.64 | 0.79 | 0.43 | -0.35 | 1.00 | - | - | - | - | |
| Benzene | 0.50 | 0.15 | 0.42 | 0.19 | 0.21 | 0.25 | 0.61 | 1.00 | - | - | - | - |
| $C_3H_6O_2$ | 0.76 | 0.35 | 0.62 | 0.43 | 0.12 | *-0.03* | 0.69 | 0.64 | 1.00 | - | - | - |
| Temperature | -0.81 | -0.35 | -0.77 | -0.53 | 0.26 | 0.10 | -0.58 | -0.40 | -0.68 | 1.00 | - | - |
| Radiation | *-0.07* | *-0.04* | -0.09 | -0.06 | 0.29 | -0.07 | *0.01* | -0.11 | -0.10 | 0.33 | 1.00 | - |
| Ozone | 0.74 | 0.70 | 0.63 | 0.79 | 0.14 | -0.26 | 0.72 | 0.31 | 0.56 | -0.64 | -0.23 | 1.00 |

a All correlations, apart from the numbers typed in italics, have linear regression p-values below 0.01.

[Figure]

**Fig. S1.** Wind Rose for mean wind speed at  5 min time resolution for (a) all seasons, (b) spring, (c) summer, and (d) autumn. The y-axis represents the percent frequency of wind direction in percent and the colors indicate mean wind speed in m s⁻¹. The seasons follow the selection outlined in Table S1.

90

[Figure]

**Fig. S2.** Time series meteorological parameters **(a)** snow depth, **(b)** radiation, **(c)** relative humidity (RH), **(d)** temperature,  **(e)** wind speed, and **(f)** wind direction during the entire measurement period.

[Figure]

**Fig. S4.** Satellite images from Sentinel 1-B, delivered by the University of Dundee, Scotland and NASA's Goddard Space Flight Center; **(a)** May 1st **(b)** May 2nd **(c)** May 3rd **(d)** May 4th **(e)** May 5th **(f)** May 6th. The presence of open leads can be seen southwest of VRS at approx. 79° 30′ N and 12° W.

[Figure]

110 **Fig. S5.** Satellite images from Sentinel 1-B, delivered by the University of Dundee, Scotland and NASA's Goddard Space Flight Center; **(a)** May 14th **(b)** May 15th **(c)** May 16th **(d)** May 17th **(c)** May 18th. The presence of open leads can be seen northeast of VRS at approx. 81° 50´ N and 10° W as well as southwest of VRS at approx. 81° N and 12° W.

115

[Figure]

**Fig. S3.** Diurnal profile for the spring (April–May) of **(a)** formaldehyde, **(b)** acetonitrile, **(c)** formic acid, **(d)** acetone, **(e)** acetic acid, **(f)** DMS, **(g)** MEK, **(h)** C$_3$H$_6$O$_2$, **(i)** benzene, **(j)** C$_5$H$_8$O. Data were averaged to hourly medians. The blue dotted lines represent the 95 % confidence interval.

[Figure]

**Fig. S4.** Diurnal profile for the summer (June–August) of **(a)** formaldehyde, **(b)** acetonitrile, **(c)** formic acid, **(d)** acetone, **(e)** acetic acid, **(f)** DMS, **(g)** MEK, **(h)** $C_3H_6O_2$, **(i)** benzene, **(j)** $C_5H_8O$. Data were averaged to hourly medians. The blue dotted lines represent the 95 % confidence interval.

125

[Figure]

**Fig. S5.** Diurnal profile for the autumn (September–October) of **(a)** formaldehyde, **(b)** acetonitrile, **(c)** formic acid, **(d)** acetone, **(e)** acetic acid, **(f)** DMS, **(g)** MEK, **(h)** $C_3H_6O_2$, **(i)** benzene, **(j)** $C_5H_8O$. Data were averaged to hourly medians. The blue dotted lines represent the 95 % confidence interval.

135

[Figure]

**Fig. S6.** HYSPLIT back trajectory analysis for **(a)** May 2nd — 6th **(b)** May 16th 20th arriving at 100 m above ground level extending 72 hours backward in time.  The colored trajectories represent a new trajectory started every 24 hours after the last day of each period until the first day, in descending order the trajectories are red (last day), blue (fourth day), green (third day), light blue (second day), and purple (first day).

140

[Figure]

**Fig. S77.** The ratio of $Q_{true}$ to $Q_{theo}$ versus the number of factors for the PMF analysis.

145

[Figure]

[Figure]

**Fig.** S8. Conditional probability function roses for **(a)** Biomass Burning Factor, **(b)** Marine Cryosphere Factor,  **(c)** Background Factor, and **(d)** Arctic Haze Factor.